# Degradation of blue-phosphorescent organic light-emitting devices involves exciton-induced generation of polaron pair within emitting layers

Sinheui Kim[1], Hye Jin Bae[2], Sangho Park[2], Wook Kim[3], Joonghyuk Kim[2], Jong Soo Kim[2], Yongsik Jung[2], Soohwan Sul[2], Soo-Ghang Ihn[2], Changho Noh[2], Sunghan Kim[2] & Youngmin You [1]

Degradation of organic materials is responsible for the short operation lifetimes of organic light-emitting devices, but the mechanism by which such degradation is initiated has yet to be fully established. Here we report a new mechanism for degradation of emitting layers in blue-phosphorescent devices. We investigate binary mixtures of a wide bandgap host and a series of novel Ir(III) complex dopants having *N*-heterocyclocarbenic ligands. Our mechanistic study reveals the charge-neutral generation of polaron pairs (radical ion pairs) by electron transfer from the dopant to host excitons. Annihilation of the radical ion pair occurs by charge recombination, with such annihilation competing with bond scission. Device lifetime correlates linearly with the rate constant for the annihilation of the radical ion pair. Our findings demonstrate the importance of controlling exciton-induced electron transfer, and provide novel strategies to design materials for long-lifetime blue electrophosphorescence devices.

[1] Division of Chemical Engineering and Materials Science, Ewha Womans University, Seoul 03760, Republic of Korea. [2] Samsung Advanced Institute of Technology, Samsung Electronics Co., Ltd., Suwon-si, Gyeonggi-do 16678, Republic of Korea. [3] Department of Electronic Materials, Samsung SDI Co., Ltd., Suwon-si, Gyeonggi-do 16678, Republic of Korea. Correspondence and requests for materials should be addressed to S.-G.I. (email: sg.ihn@samsung.com) or to S.K. (email: shan0819.kim@samsung.co) or to Y.Y. (email: odds2@ewha.ac.kr)

Extensive research has enabled organic light-emitting devices (OLEDs) to outperform conventional displays in various commercial applications. One widely recognized remaining challenge, however, is to improve device lifetime. In particular, the short operation time of devices that emit blue light are significant impediments to exploiting the full potential of OLEDs.

Growing evidence has ascribed the short device lifetime to an accumulation of chemical defects resulting from irreversible degradation of organic materials during device operation[1–4]. The mechanism by which organic materials degrade involves radical species as byproducts and key reaction intermediates[5–10]. Once formed, the open-shell radical species deteriorates device performance by trapping charge carriers or by quenching light emission through exciton–polaron annihilation or nonradiative recombination[11–21]. Previous studies established exciton-localized homolysis and imbalance in charge carrier concentrations to be responsible for the generation of the radical species[16, 18, 22–25]. Molecules containing weak covalent bonds undergo homolysis in the excited state, producing neutral radical species[26–30]. Charged radical species (e.g., polarons) are generated inevitably via charge carrier injection and transport during normal device operation. Efforts have thus been made to minimize the population of radical species within emitting layers by decreasing the density of excitons or by suppressing the buildup of charge[19, 31–36].

Despite these advances, there has been a limited understanding of the fundamental reason for the significant instability of devices that emit blue light. Note that an intermolecular pathway for radical generation has been overlooked. It is possible that the exciton is reductively quenched by electron transfer from a nearby molecule with a shallow oxidation potential to form a radical ion pair (i.e., a polaron pair), due to the positive driving force of the electron transfer (Fig. 1). Such radical ion pair would rapidly undergo charge recombination to restore the original neutral states, but its labile nature can facilitate degradation of both the host and dopant materials. This mechanism may explain the greater instability of devices that emit blue light than those that emit green and red light, because it predicts faster formation and slower annihilation of the radical ion pair in blue emission layers (vide infra). Therefore, it is envisioned that studies of intermolecular electron transfer can lead to an ability to control on the radical species responsible for intrinsic degradation of emitting layers.

In the current research, we investigate a charge-neutral, exciton-induced generation of radical ion pairs between a wide bandgap host and blue-phosphorescent Ir(III) complex dopants. The radical ions and their annihilation processes in the host–dopant binary system are directly monitored for the first time, with employing a variety of chemical techniques. The rate constants for charge recombination by back electron transfer ($k_{BeT}$) are determined through second-order kinetics analyses. Multilayer OLEDs are fabricated, and a linear correlation is found between $k_{BeT}$ and device lifetime. The study reveals the importance of controlling the electrochemical potentials of a host exciton and its dopant for achieving long device lifetimes.

## Results

**Synthesis of materials and thermodynamic analyses of electron transfer.** A series of homoleptic triscyclometalated Ir(III) complexes having $N$-heterocyclocarbenic (NHC) ligands (Ir1–Ir4) were newly synthesized and employed as blue-phosphorescent dopants. 3,3′-Biscarbazolyl-5-cyanobiphenyl (H) was used as a wide-bandgap host material[37]. Chemical structures of the Ir dopants and H are depicted in Fig. 2a. Synthetic details and structural characterization data for the compounds are summarized in Methods. Structures of the benzimidazole-based (Ir1 and Ir2) and imidazopyrazine-based (Ir3 and Ir4) NHC ligands were systematically varied to tailor the electrochemical potentials of the complexes. This synthetic control was indeed demonstrated by an observation of a shift of the reversible Ir(IV/III) redox potentials (0.81−1.12 V vs. standard calomel electrode (SCE); Fig. 2b). H displayed irreversible one-electron oxidation and reduction processes at 1.50 V ($E_{ox}$) and −1.82 V ($E_{red}$) vs. SCE, respectively (Supplementary Fig. 1). An optical bandgap energy ($\Delta E_g$) as large as 3.10 eV was determined, which enabled calculation of the excited-state oxidation ($E^\star_{ox}$) and reduction ($E^\star_{red}$) potentials of H according to the relationships $E^\star_{ox} = E_{ox} - \Delta E_g$ and $E^\star_{red} = E_{red} + \Delta E_g$. In this way, $E^\star_{ox}$ and $E^\star_{red}$ of H were calculated to be −1.60 and 1.28 V vs. SCE, respectively. Comparison of the $E^\star_{red}$ value of H with the oxidation potentials of the Ir dopants revealed positive driving forces ($-\Delta G_{eT}$) in the range 0.16−0.47 eV for the reductive quenching of an H exciton through forward one-electron transfer from the dopant, where $-\Delta G_{eT}$ was calculated from the equation $-\Delta G_{eT} = e\cdot[E_{ox}(Ir) - E^\star_{red}(H)]$ (in this equation, $e$ is the elementary charge of an electron) (Fig. 2d). This thermodynamic consideration predicted the generation of pairs each consisting of a radical anion of H with a radical cation of Ir, i.e., [H•−Ir•+]. Charge recombination can occur within the radical ion pair through back electron transfer from H•− to Ir•+. The driving force for back electron transfer ($-\Delta G_{BeT}$) can be estimated from the equation $-\Delta G_{BeT} = e\cdot[E_{red}(H) - E_{ox}(Ir)]$. The $-\Delta G_{BeT}$ values were calculated to be as large as 2.63−2.94 eV (Fig. 2e), suggesting rapid charge recombination. Photophysical and electrochemical parameters for the H and Ir compounds are compiled in Table 1.

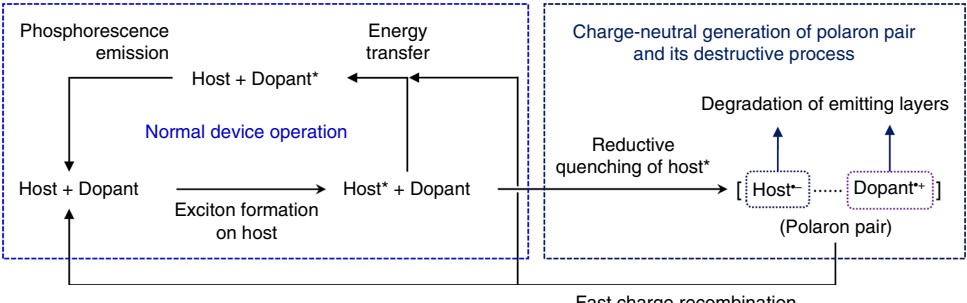

**Fig. 1** Exciton-induced generation of polaron pairs. Electronic processes for the charge-neutral, exciton-mediated generation of radical ion pairs within an emitting layer consisting of a host and a dopant. Direct formation of a dopant exciton (e.g., dopant*) is not included, because intermolecular electron transfer between a host and a dopant exciton is usually forbidden due to negative driving forces

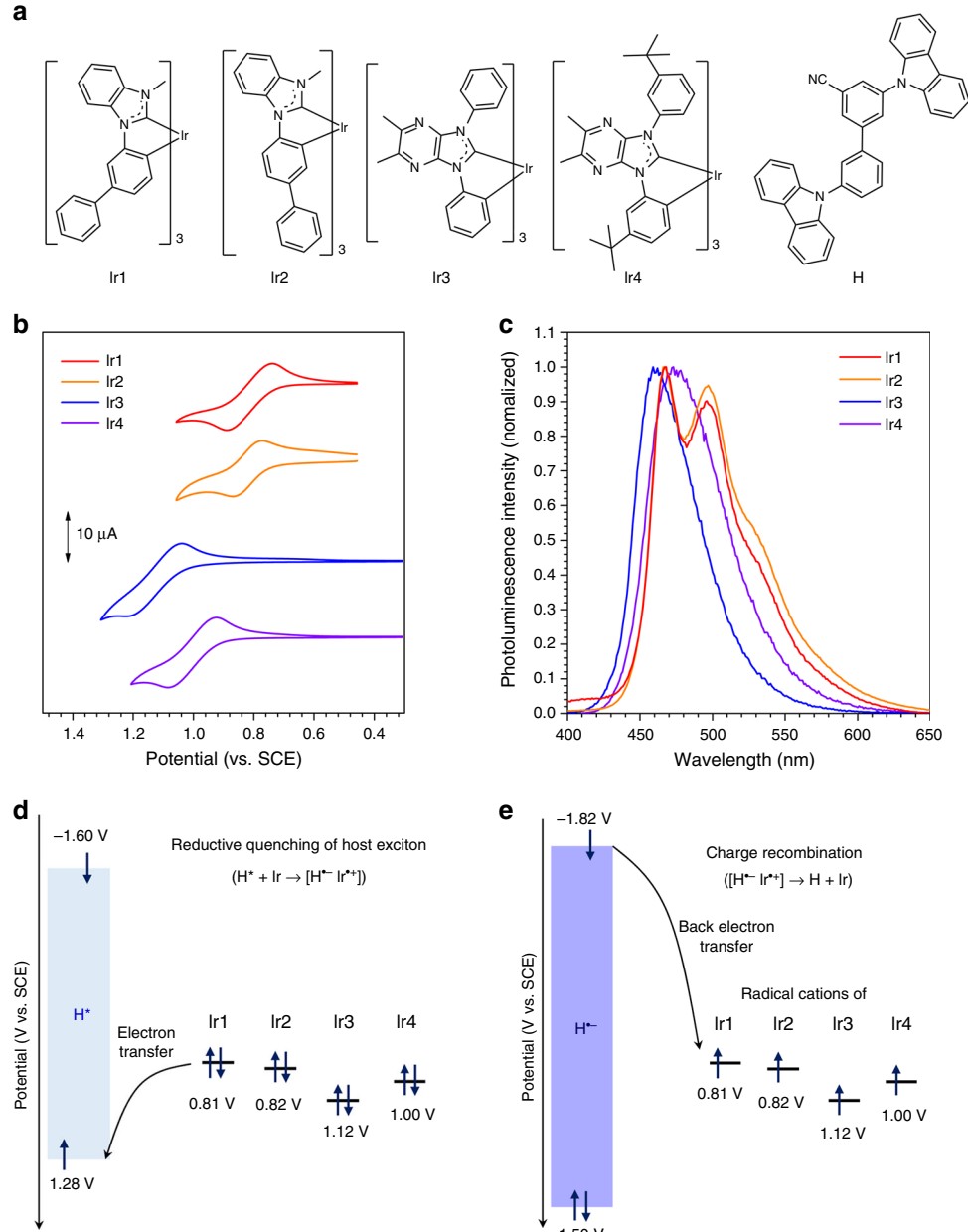

**Fig. 2** Prediction of the formation of radical ion pairs. **a** Chemical structures of the blue-phosphorescent dopants (Ir1–Ir4) and the host molecule (H). **b** Cyclic voltammograms of the Ir dopants. Conditions: a standard three-electrode cell assembly consisting of a Pt disc working electrode, a Pt wire counter electrode, and an Ag/AgNO₃ pseudo reference electrode were employed. Solutions of 2.0 mM compounds in THF were deaerated prior to taking the measurements. **c** Photoluminescence spectra of 10 μM Ir dopant solutions (THF, deaerated). Photoexcitation wavelengths used were 341 nm (for Ir1), 340 nm (for Ir2), 396 nm (for Ir3), and 400 nm (for Ir4). **d** Comparison of the electrochemical potentials of the host and dopants involved in reductive quenching of the host exciton. **e** Comparison of the electrochemical potentials of the host and dopants involved in charge recombination within a pair of the radical anion of the host and the radical cation of the dopant

**Formation and annihilation of radical ion pairs.** Steady-state and transient spectroscopic techniques were employed to monitor the electron-transfer processes. Fluorescence quenching experiments were performed upon excitation of the host for a solution of H (100 μM, Ar-saturated THF) with the addition of the Ir3 dopant (Fig. 3a and Supplementary Fig. 2 for the other Ir dopants). A nonlinear least-squares fit of the results to the Stern–Volmer equation yielded a rate constant for bimolecular quenching greater than $10^{12}\,M^{-1}\,s^{-1}$. Although energy transfer to Ir3 cannot be excluded, the observation that sensitized phosphorescence of Ir3 due to energy transfer from H decreased with increasing equimolar concentrations of Ir3 and H suggested a

concentration-dependent generation of non-emissive $[H^{\bullet-}Ir^{\bullet+}]$ (Fig. 3b).

Photoinduced electron paramagnetic resonance (EPR) spectroscopy provided direct evidence for the formation of radical species. As shown in Fig. 3c, 1.0 mM H, 1.0 mM Ir3, and a mixture of 1.0 mM H and 1.0 mM Ir3 (Ar-saturated THF) did not display any apparent paramagnetic signals in the dark. Photo-irradiation of the binary mixture of H and Ir3 produced an increase in paramagnetic signals at a g value of 2.012, typical of a free radical. This peak was accompanied by rhombic signals presumably due to an Ir(IV) species of Ir•+. These peaks disappeared upon cessation of photoirradiation, which indicated

**Table 1 Photophysical and electrochemical data for the Ir dopant and H host materials**

| | $\lambda_{em}$ (nm)[a] | $\tau_{obs}$(µs)[b] | PLQY[c] | $E_{ox}$(V vs SCE)[d] | $E_{red}$(V vs SCE)[e] | $E^\star_{ox}$(V vs SCE)[f] | $E^\star_{red}$(V vs SCE)[g] |
|---|---|---|---|---|---|---|---|
| Ir1 | 467 | 2.7 | 0.13 | 0.81 (r) | _[h] | −1.85 | ND[i] |
| Ir2 | 467 | 2.0 | 0.25 | 0.82 (r) | _[h] | −1.84 | ND[i] |
| Ir3 | 460 | 3.8 | 0.28 | 1.12 (r) | _[h] | −1.59 | ND[i] |
| Ir4 | 474 | 2.3 | 0.50 | 1.10 (r) | _[h] | −1.63 | ND[i] |
| H | 400 | 0.0075 | ND[i] | 1.50 (ir) | −1.82 (ir) | −1.60 | 1.28 |

*r* reversible, *ir* irreversible
[a] 10 µM in deaerated THF, 298 K
[b] Photoluminescence lifetimes determined employing time-correlated single-photon-counting techniques, after picosecond pulsed laser excitation at 377 nm. The measurements were taken for deaerated THF solutions containing 50 µM Ir dopant or 50 µM H host
[c] Photoluminescence quantum yields of the Ir dopants measured for mCBP films (50 nm) molecularly dispersed with 5 wt % Ir
[d] Oxidation potentials
[e] Reduction potentials. Cyclic (scan rate = 100 mV s⁻¹), differential pulse (scan rate = 4 mV s⁻¹) and second harmonic alternating current (scan rate = 25 mV s⁻¹) voltammetry experiments were performed to determine the potentials. The electrochemical measurements were taken for deaerated THF solutions of 2.0 mM Ir or 2.0 mM H employing a three-electrode cell assembly consisting of a Pt disc working electrode, a Pt wire counter electrode, and an Ag/AgNO3 pseudo reference electrode
[f] Excited-state oxidation potentials
[g] Excited-state reduction potentials
[h] Not observed before solvent breakdown
[i] Not determined

that charge recombination occurred. Taken together, the results revealed that a radical ion species formed through rapid electron transfer.

To monitor the generation and annihilation of the radical ion species, transient absorption spectra were acquired for deaerated THF solutions containing 150 µM Ir3 and 3.0 mM H. As shown in Fig. 4a, weak positive absorption signals were observed in the NIR region of the spectra, although the spectra were dominated by the emission of Ir3. The positive signals were not seen for solutions containing only Ir3 or only H, which indicated electron-transfer species (e.g., radical ion pairs) to be responsible for the NIR signals. Comparison of the transient absorption spectra with simulated electronic transition energies of $H^{\bullet-}$ (the magenta curve in Fig. 4b) and $Ir3^{\bullet+}$ (the blue curve in Fig. 4b) (TD–CAM–B3LYP/LANL2DZ:6–311G(d,p)//CAM–B3LYP/LANL2DZ:6–311G(d,p)) indicated that the NIR spectra were due to $Ir3^{\bullet+}$. This assignment was further corroborated by the spectroelectrochemical measurement taken for Ir3 (2.0 mM, deaerated THF) during oxidative electrolysis at 1.30 V vs. SCE (the black curve in Fig. 4b). Here, spectral signatures of $H^{\bullet-}$ were not observed, presumably due to the instability of the one-electron-reduced state, as suggested from the irreversible reduction process.

Charge recombination by back electron transfer from $H^{\bullet-}$ to $Ir3^{\bullet+}$ was monitored by recording the decay of the transient absorption signal of $Ir3^{\bullet+}$ at a wavelength of 1100 nm. Its $k_{BeT}$ value was determined to be $1.5 \times 10^{11}$ M⁻¹ s⁻¹ according to a second-order kinetics analysis (Fig. 4c); this analysis employed a molar absorbance value of $Ir3^{\bullet+}$ ($\varepsilon = 2170$ M⁻¹ cm⁻¹), which was determined from the spectroelectrochemical measurements. The $k_{BeT}$ values determined for the other Ir dopants were $4.5 \times 10^{10}$ M⁻¹ s⁻¹ (for Ir1), $3.0 \times 10^{10}$ M⁻¹ s⁻¹ (for Ir2), and $2.8 \times 10^{11}$ M⁻¹ s⁻¹ (for Ir4). The $k_{BeT}$ values corresponded well to the theoretical curves obtained from the classical Marcus theory for adiabatic outer-sphere electron transfer, with the reorganization energies in the ranges 1.7–1.9 eV and 2.2–2.3 eV for the benzimidazole-based NHC Ir(III) complexes (Ir1 and Ir2) and the imidazopyrazine-based NHC Ir(III) complexes (Ir3 and Ir4), respectively (Fig. 4d). Note that $k_{BeT}$ was observed to increase as the value of $-\Delta G_{BeT}$ became smaller, which revealed that the charge recombination was located in the Marcus-inverted region of electron transfer[38]. Since more rapid charge recombination is beneficial for suppression of irreversible degradation from radical ion species, this result provided two valuable approaches for improving device lifetimes: decreasing $-\Delta G_{BeT}$ and increasing the reorganization energy. The former approach may involve shifting the

$E_{ox}$ of a dopant cathodically by increasing the electron density of the NHC ligand. The latter approach requires further understanding about reorganization processes, but potentially involves the use of charge-delocalizing ligand frameworks.

**Degradation from radical ion pairs**. It was found during the laser flash photolysis experiments that the binary mixture of H and Ir4 underwent irreversible coloration, which suggested that these materials became degraded. UV–vis absorption changes were observed for a mixture of H and Ir4 during steady-state photolysis upon irradiation using a Xenon lamp (300 W). The UV–vis absorption spectrum of Ir4 showed a characteristic intense band at 390 nm due to an intraligand charge transfer (ILCT) transition localized within the NHC ligand. The absorbance of the ILCT transition band decreased during the photolysis (Fig. 5a and Supplementary Fig. 3), which indicated that the NHC ligand fragmented. A similar bleaching behavior was observed for vacuum-evaporated films of H containing 15 wt % Ir4 (Fig. 5b). As expected, such an absorption change was absent for redox-innocent poly(methyl methacrylate) films doped with 15 wt % Ir4 (Fig. 5c).

Liquid chromatograms obtained for the photolyzed samples revealed the formation of byproducts (Fig. 5d, e). Electrospray ionization mass spectra of the byproducts displayed peaks corresponding to fragments due to cleavage of the bond connecting the imidazopyrazine and phenyl moieties in the NHC ligands in Ir4 (Fig. 5f). Such fragmentation products were found for Ir4 after its oxidative electrolysis at 1.25 V vs. SCE (Fig. 5g). Analyses of the degradation products of Ir1–Ir3 revealed fragmentation behaviors identical to Ir4 (Supplementary Fig. 4). The degradation was also observed for vacuum-evaporated films of H doped with 15 wt % Ir during photoirradiation (Supplementary Fig. 5). The results collectively indicated that the Ir dopants degraded upon undergoing one-electron oxidation. Tang and co-workers suggested a radical cation of a blue-phosphorescent Ir(III) complex to be unstable[8]. In addition to observing the degradation of dopants, we were able to observe byproducts of the host. These byproducts mainly originated from scission of C–N bonds between biphenyl and carbazole groups of $H^{\bullet-}$ (Supplementary Fig. 4 and 5). Such bond scission was also reported for host materials having similar biscarbazolylbiphenyl frameworks[14]. On the basis of the results, we conclude that unstable radical ion species are generated intermolecularly between a host and a dopant molecules, even without the charge carrier injection, and that these radical ions are responsible for the irreversible degradation.

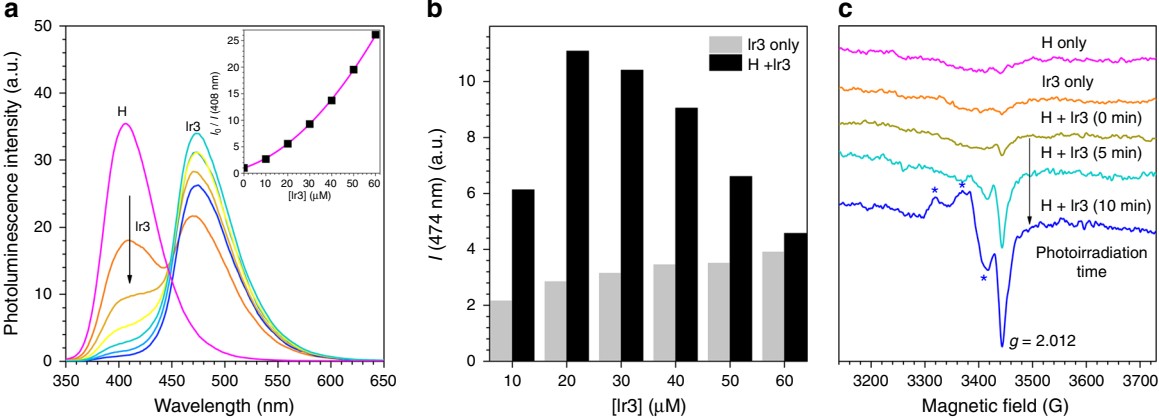

**Fig. 3** Formation of radical ion pairs. **a** Photoluminescence spectra of 100 μM H with added Ir3 (0–60 μM). Photoexcitation wavelength used was 300 nm. The inset figure is a Stern–Volmer analysis of the fluorescence of H in the absence ($I_O$) and presence ($I$) of Ir3. The value of the intensity of the host fluorescence was corrected by considering the absorbance of the dopant, following the relationship $I = I_{obs} \times (Abs/Abs_O) \times 1/(1 - 10^{-Abs})$ where $I_{obs}$, $Abs$, and $Abs_O$ are the observed fluorescence intensity, the absorbance at 300 nm in the presence of Ir3, and the absorbance at 300 nm in the absence of Ir3, respectively. The $I_O/I$ values were fit to the Stern–Volmer equation $I_O/I = (1 + K_a \cdot [Ir3]) \times (1 + k_q \cdot \tau_O \cdot [Ir3])$. In this equation, $K_a$, $k_q$, $\tau_O$, and [Ir3] are the association constant, the quenching constant, the fluorescence lifetime in the absence of Ir3 (3.8 ns), and the molar concentration of Ir3, respectively. The Stern–Volmer analysis yielded $k_q$ and $K_a$ to be >$10^{12}$ M$^{-1}$ s$^{-1}$ and $6.8 \times 10^3$ M$^{-1}$, respectively. The non-negligible contribution of the static quenching (i.e., the $(1 + K_a \cdot [Ir3])$ term in the Stern–Volmer equation) may indicate the existence of excited-state interactions between H and Ir3. Stern–Volmer analyses for other Ir dopants are shown in Supplementary Fig. 2. **b** Phosphorescence intensities of 10, 20, 30, 40, 50, and 60 μM Ir3 (deaerated THF) upon photoexcitation at 300 nm in the absence (gray bars) and presence (black bars) of equimolar concentrations of H. The decrease in the difference between the gray and black bars corresponded to the formation of radical ion pairs. **c** Photoinduced EPR spectra of Ar-saturated THF solutions of 1.0 mM H, 1.0 mM Ir3, and a mixture of 1.0 mM H and 1.0 mM Ir3 in the absence and presence of photoirradiation. Peaks marked with asterisks may correspond to the rhombic signals due to an Ir(IV) species of the radical cation of Ir3

**Correlation between device lifetime and the kinetics of radical ion pair.** Having established the degradation mechanism, we attempted to correlate the electron-transfer kinetics with device lifetime. Multilayer blue-phosphorescent OLEDs were fabricated using vacuum evaporation techniques with an ITO/HAT–CN/NPB/mCP/emitting layer/DBFPO/Liq/Al device configuration (Fig. 6a). HAT–CN (1, 4, 5, 8, 9, 11-hexaazatriphenylenehexacarbonitrile), NPB (N,N-di(1-naphthyl)-N,N′-diphenyl-(1,1′-biphenyl)-4,4′-diamine), mCP (1,3-bis(N-carbazolyl)benzene), DBFPO (2,8-bis(diphenylphosphineoxide)-dibenzofuran), and Liq (lithium quinolinate) served as a hole-injection layer, a hole-transporting layer, an electron-blocking layer, an electron-transporting and hole-blocking layer, and a buffer layer, respectively. The emitting layer consisted of a 30-nm-thick film of H that was molecularly doped with 10 or 20 wt % Ir dopant. The electroluminescence spectra (Fig. 6b) were identical to the photoluminescence spectra (Fig. 1c), excluding the contribution of any bimolecular species, such as exciplex, to the electroluminescence emission. An external quantum efficiency (EQE) as high as 17.8% was recorded at 500 cd m$^{-2}$ for a 20 wt % Ir4 device, and the efficiency decreased in the order Ir3, Ir2, and Ir1 (Fig. 6d). While we found this trend to be consistent with the photoluminescence quantum yields (PLQYs), the significantly larger EQE values for Ir4 and Ir3 than those for the other two dopants may have been due to regeneration of dopant excitons through back electron transfer from H$^{\bullet -}$ to Ir$^{\bullet +}$. In the case of Ir3 and Ir4, their $-\Delta G_{BeT}$ values (Ir3, 2.94 eV; Ir4, 2.82 eV) were indeed calculated to be greater than their triplet exciton energy ($\Delta E_T$) values (Ir3, 2.70 eV; Ir4, 2.62 eV), allowing for the exciton recovery. Such exciton recovery, however, was inferred to be disallowed for Ir1 and Ir2, since their $-\Delta G_{BeT}$ values (Ir1, 2.63 eV; Ir2, 2.64 eV) were calculated to be less than their $\Delta E_T$ values (Ir1, 2.66 eV; Ir2, 2.66 eV). A summary of obtained electroluminescence data for the H:Ir devices is shown in Supplementary Table 2.

To correlate device lifetime with the electron-transfer behavior, luminance decays of the H:Ir devices were recorded during operation in a constant current driving mode with an initial current value defined at a luminance of 500 cd m$^{-2}$. Figure 6e depicts the luminance profiles of the devices, each as a function of operation time. $LT_{70}$, which corresponds to the operation time when the luminance decreases to 70% of its initial value, approached 93.05 h for the 20 wt % Ir4 device, and decreased in the order Ir4, Ir3 (19.70 h), Ir2 (4.11 h), and Ir1 (1.88 h). The dissimilarity between this $LT_{70}$ order and the order of the triplet state energy value of the Ir dopants was interpreted as excluding exciton-localized homolysis of the dopant from being a dominant degradation pathway. It is worth noting that the $LT_{70}$ value was observed to increase in proportion with $k_{BeT}$ (Fig. 6f). Photoluminescence stability of the films of H:Ir also followed this trend, indicating that the intermolecular electron-transfer reactions, not different current levels in devices, were responsible for device lifetime (Supplementary Fig. 6). These results provided direct evidence for acceleration of charge recombination being crucial for device longevity. Finally, validity of our mechanism was further examined by comparing the $LT_{70}$ values with those of devices having mCBP (3,3′-bis(9-carbazolyl)biphenyl) in place of H as a host material. It is predicted that radical ion pairs are more accumulated in the mCBP:Ir layers than the H:Ir layers, due to the more positive $E^*_{red}$ (1.88 V vs. SCE; c.f., $E^*_{red}$ of H = 1.28 V vs. SCE) and similar $E_{red}$ (−1.79 V vs. SCE; c.f., $E_{red}$ of H = −1.82 V vs. SCE) values of mCBP. Indeed, devices of mCBP exhibited poor longevity consistently for all the Ir dopants, as quantitated from shorter $LT_{70}$ values (Supplementary Table 3).

**Discussion**

We have investigated exciton-induced generation of radical ion pairs between a wide bandgap energy host and blue-phosphorescent Ir(III) complexes having NHC ligands. Spectroscopic techniques were employed to detect charge-neutral

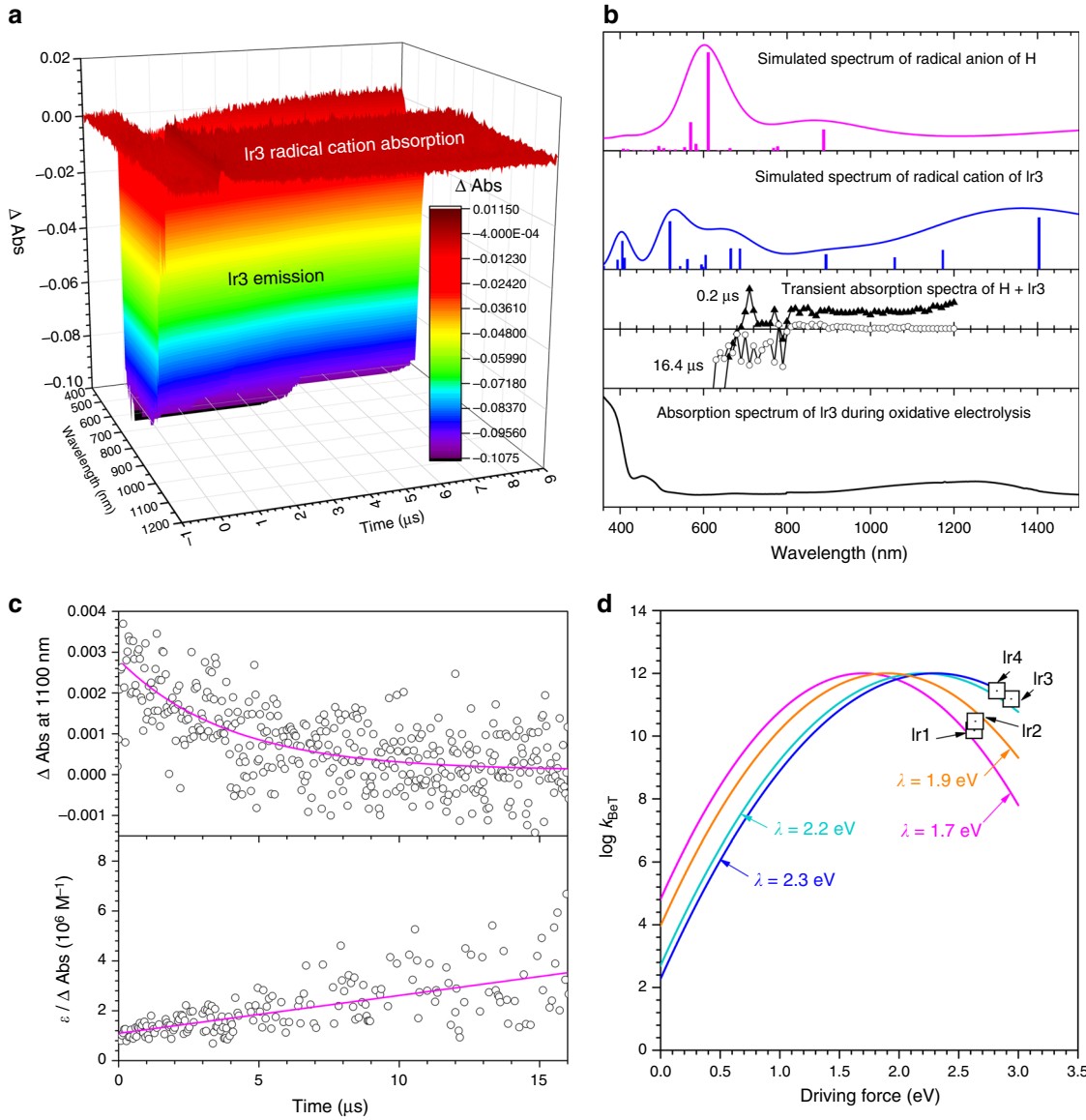

**Fig. 4** Annihilation of radical ion pairs. **a** Transient absorption spectra of a THF solution containing 150 µM Ir3 and 3.0 mM H after nanosecond pulsed photoexcitation. The negative and positive absorbance were due to the emission of Ir3 and the generation of Ir3•+, respectively. **b** Comparisons of the simulated (TD−B3LYP) electronic transition spectra of the radical anion of H (magenta curve) and the radical cation of Ir3 (blue curve) with the transient absorption spectra obtained at delay times of 0.2 µs (black triangles) and 16.4 µs (empty circles), and the absorption spectrum of Ir3 obtained during oxidative electrolysis at 1.30 V vs. SCE (black curve). **c** Decay traces (top panel) and a second-order plot (bottom panel) of the transient absorption signals observed at a wavelength of 1100 nm. **d** Plot of the rate constant for back electron transfer for charge recombination ($k_{BeT}$, black squares) as a function of the driving force and the Marcus plots for outer-sphere electron transfer calculated for the reorganization energies ($\lambda$) at 1.7 eV (magenta), 1.9 eV (orange), 2.2 eV (turquoise), and 2.3 eV (blue). The Marcus plots were constructed from the equation, $k_{BeT} = Z \exp[-(\Delta G_{BeT} + \lambda)^2/4\lambda k_B T]$. In this equation, $Z$, $k_B$, and $T$ are the collisional frequency taken as $1.0 \times 10^{12}$ M$^{-1}$ s$^{-1}$, the Boltzmann constant, and absolute temperature (298 K), respectively

production of a pair of the radical cation of the dopant and the radical anion of the host. In particular, photoinduced EPR experiments and nanosecond laser flash photolysis provided direct evidence for the formation of the radical cation of the dopant molecule. Charge recombination kinetics within the radical ion pair was analyzed by employing the second-order kinetics model. Charge recombination was found to occur in the Marcus-inverted region of electron transfer. Analyses of the radical ion-mediated degradation products revealed the occurrence of oxidative cleavage of the dopant and reductive degradation of the host. Finally, a strong linear proportionality was found between the device lifetime and the rate constant for charge recombination. This finding is valuable, because it provides reasonable explanations for the poor stability of OLEDs that emit specifically blue light. Here,

large driving forces for electron transfer ($-\Delta G_{eT}$) facilitate more rapid formation of radical ion pairs, but much larger driving forces for back electron transfer ($-\Delta G_{BeT}$) impede charge recombination occurring in the Marcus-inverted region of electron transfer. The combined effect is a longer lifetime of radical ion pairs in an emitting layer of a blue-phosphorescent OLED than those of green-phosphorescent and red-phosphorescent OLEDs. It should be noted that accumulated levels of the radical ion species are minimal, because the forward electron transfer competes with energy transfer to a dopant, and, also, rapid charge recombination occurs. Overall, our mechanistic study revealed that exercising the electrochemical control to minimize the density and lifetime of radical ion pairs is crucial for realizing long operation lifetimes of OLEDs.

## Methods

**General procedures**. Chemicals were purchased from commercial suppliers, and used without further purification. $^1$H and $^{13}$C{$^1$H} NMR spectra were recorded on a Bruker, AVANCE III 600 spectrometer, with employing $CD_2Cl_2$ as the solvent. Chemical shifts were referenced to the peaks corresponding to residual solvents. Mass spectra (MALDI−TOF) were recorded using a Bruker, Ultraflex III TOF/TOF 200 spectrometer. The host molecule (H) was prepared following the method

reported previously[1]. Purity of the Ir dopants was examined by performing high performance liquid chromatography (HPLC) analyses. A Waters, Alliance e2695 was employed for the HPLC measurements (conditions: column, Phenomenex (2.6 µm, C18 100 A, 100 × 4.6 mm); eluent, acetonitrile:water = 70:30 (v/v), gradient); sampling solvent, THF; oven temperature, 40 °C; flow rate, 1.0 mL min$^{-1}$; APCI mode). LC-mass analyses were carried out to validate the HPLC results, employing a Shimadzu, LCMS−IT−TOF instrument. Elemental analyses were also performed

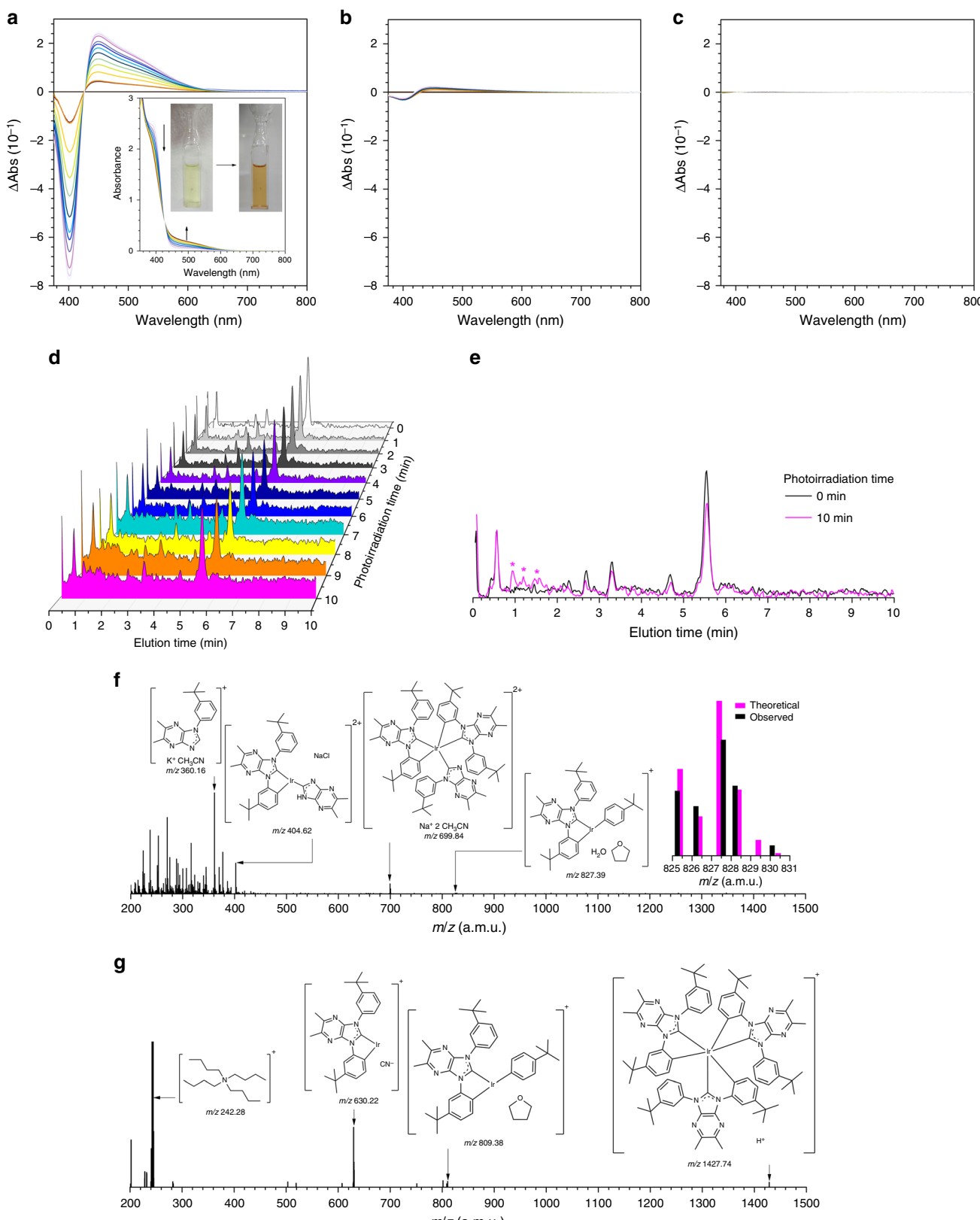

for C, H, and N on Thermo Fisher Scientific, Flash1112 or Flash2000 instruments. H was prepared following the method reported previously[37]. Organic materials for device fabrication were purchased from commercial suppliers, and purified by sublimation at $10^{-5}$ torr prior to deposition. Films were deposited under high vacuum ($<1.0 \times 10^{-6}$ torr) onto a pre-cleaned glass substrate (1 cm × 1 cm). Poly (methyl methacrylate) (PMMA, Mw ~120,000, Sigma–Aldrich) films doped with the Ir dopants (15 wt %) were dissolved in 1, 2-dichoroethane (5 wt % total solute relative to solution). The solution was sonicated for 30 min, and passed through a membrane filter (pore size = 8.0 μm). An aliquot of the polymer solution was placed onto a pre-cleaned glass substrate, and was spin-casted with employing an EPLEX, SPIN-1200D spin coater. Spectrophotometric grade THF stored under an insert atmosphere was used for spectroscopic and electrochemical measurements.

**Synthesis of Ir1.** A DMF solution containing 1-((1,1′-biphenyl-3-yl)-3-methyl-1H-benzo[d]imidazol-3-ium iodide (2.50 g, 6.06 mmol), silver(I) oxide (0.70 g, 3.03 mmol), and $IrCl_3 \cdot 3H_2O$ (0.55 g, 1.84 mmol) was refluxed for 16 h under an Ar atmosphere. After cooling to room temperature, the solvent was removed in vacuo. The crude product was purified by column chromatography on silica gel using dichloromethane as an eluent. Recrystallization in dichloromethane and n-hexane gave Ir1 as a pale yellow powder in a 57% yield (1.10 g). [1]H NMR (600 MHz, $CD_2Cl_2$): δ 8.30 (d, J = 8.2 Hz, 3 H), 8.17 (s, 3 H), 7.70 (d, J = 7.7 Hz, 6 H), 7.44 (t, J = 7.6 Hz, 6 H), 7.42–7.36 (m, 3 H), 7.32 (d, J = 6.3 Hz, 9 H), 7.00 (d, J = 7.6 Hz, 3 H), 6.77 (d, J = 7.5 Hz, 3 H), 3.39 (s, 9 H); [13]C{[1]H} NMR (150 MHz, $CD_2Cl_2$): δ 190.09, 150.41, 148.71, 143.03, 137.56, 136.97, 134.86, 133.15, 129.28, 127.16, 126.74, 123.77, 123.45, 122.65, 111.89, 111.50, 110.57, 34.25; MALDI–TOF/MS (m/z): $[M]^+$ calcd. for $C_{60}H_{45}IrN_6$, 1042.33; found, 1042.49; analysis (calcd., found for $C_{60}H_{45}IrN_6$): C (69.14, 69.16), H (4.35, 4.34), N (8.06, 8.06).

**Synthesis of Ir2.** The method identical to that for Ir1 was employed, except the use of 1-((1,1′-biphenyl-4-yl)-3-methyl-1H-benzo[d]imidazol-3-ium iodide (3.0 g, 7.28 mmol) in place of 1-((1,1′-biphenyl-3-yl)-3-methyl-1H-benzo[d]imidazol-3-ium iodide. A pale yellow powder (40%, 0.93 g). [1]H NMR (600 MHz, $CD_2Cl_2$): δ 8.21 (d, J = 8.2 Hz, 3 H), 7.96 (d, J = 8.2 Hz, 3 H), 7.39–7.29 (m, 12 H), 7.32–7.26 (m, 6 H), 7.20 (t, J = 7.6 Hz, 6 H), 7.12 (t, J = 7.3 Hz, 3 H), 7.00 (d, J = 2.2 Hz, 3 H), 3.39 (s, 9 H); [13]C{[1]H} NMR (150 MHz, $CD_2Cl_2$): δ 189.84, 149.79, 149.24, 142.99, 137.14, 136.95, 135.98, 133.13, 128.80, 127.23, 126.40, 123.34, 122.51, 120.62, 112.76, 111.80, 110.41, 34.18, 30.28; MALDI–TOF/MS (m/z): $[M]^+$ calcd. for $C_{60}H_{45}IrN_6$, 1042.33; found, 1042.56; analysis (calcd., found for $C_{60}H_{45}IrN_6$): C (69.14, 68.78), H (4.35, 4.35), N (8.06, 7.99).

**Synthesis of Ir3.** An o-xylene (75 mL) solution containing 5,6-dimethyl-1,3-diphenyl-1H-imidazo[4,5-b]pyrazin-3-ium iodide (2.7 g, 6.30 mmol), silver(I) oxide (2.00 g, 8.64 mmol), and chloro(1,5-cyclooctadiene)iridium(I) dimer (1.50 g, 2.16 mmol) was refluxed for 20 h under an Ar atmosphere. After cooling to room temperature, the solvent was removed in vacuo. The crude product was purified by column chromatography on silica gel, using dichloromethane/n-hexane (2/1, v/v) as an eluent. Recrystallization in dichloromethane/n-hexane gave Ir3 as a yellow crystal (60%, 1.03 g). [1]H NMR (600 MHz, $CD_2Cl_2$): δ 8.80 (dd, J = 7.8, 1.2 Hz, 3 H), 7.13 (td, J = 7.7, 1.5 Hz, 3 H), 6.87–6.81 (m, 3H), 6.77 (td, J = 7.2, 1.2 Hz, 3 H), 6.61 (dd, J = 7.5, 1.5 Hz, 3 H), 6.48 (br s, 12 H), 2.72 (s, 9 H), 2.43 (s, 9 H); [13]C{[1]H} NMR (150 MHz, $CD_2Cl_2$): δ 191.87, 147.64, 147.11, 145.76, 144.67, 141.13, 138.24, 136.82, 136.63, 128.84, 128.47, 128.06, 125.85, 122.24, 115.31, 22.46, 22.17; MALDI–TOF/MS (m/z): $[M]^+$ calcd. for $C_{57}H_{45}IrN_{12}$, 1090.35; found, 1090.58; analysis (calcd., found for $C_{57}H_{45}IrN_{12}$): C (62.79, 62.82), H (4.16, 4.21), N (15.42, 15.51).

**Synthesis of Ir4.** The method identical to that for Ir3 was employed, except the use of 1,3-bis(3-(tert-butyl)phenyl)-5,6-dimethyl-1H-imidazo[4,5-b]pyrazin-3-ium iodide (2.4 g, 4.68 mmol) in place of 5,6-dimethyl-1,3-diphenyl-1H-imidazo[4,5-b]pyrazin-3-ium iodide. A yellow crystal (46%, 0.77 g). [1]H NMR (600 MHz, $CD_2Cl_2$): δ 8.76 (s, 3 H), 6.83 (s, 3 H), 6.75 (dd, J = 7.8, 2.0 Hz, 3 H), 6.63 (br s, 9 H), 6.31 (d, J = 7.8 Hz, 3 H), 2.66 (s, 9 H), 2.43 (s, 9 H), 1.35 (s, 27 H), 1.18 (s, 27 H); [13]C{[1]H} NMR (150 MHz, $CD_2Cl_2$): δ 192.96, 152.28, 147.51, 144.98, 144.07, 141.25, 138.01,

136.74, 135.38, 127.72, 125.41, 125.12, 122.71, 112.46, 35.05, 34.80, 31.98, 31.49, 22.49, 22.14; MALDI–TOF/MS (m/z): $[M]^+$ calcd. for $C_{81}H_{93}IrN_{12}$, 1426.73; found, 1427.01; analysis (calcd., found for $C_{81}H_{93}IrN_{12}$): C (68.18, 68.25), H (6.57, 68.25), N (11.78, 11.81).

**Steady-state UV–vis absorption measurements.** UV–vis absorption spectra were collected on an Agilent, Cary 300 spectrophotometer. Solution samples were prepared in THF to a 10 μM concentration prior to the measurements, unless otherwise stated.

**Steady-state photoluminescence measurements.** Photoluminescence spectra were obtained using a PTI, Quanta Master 400 scanning spectrofluorimeter at 298 K. The 10 μM solutions or the films were used for the measurements. The Ir solutions were deaerated by bubbling Ar for >15 min. Photoexcitation wavelengths were 341 nm (Ir1), 340 nm (Ir2), 396 nm (Ir3), 400 nm (Ir4), and 340 nm (H). The PLQYs were determined employing an absolute PLQY measurement system (Hamamatsu, C11347-01).

**Photoluminescence lifetime measurements.** Ar-saturated 50 μM solutions in THF were used for the determination of the $\tau_{obs}$ values of the Ir dopants and H host. Photoluminescence decay traces were acquired based on time-correlated single-photon-counting (TCSPC) techniques using a FluoTime 200 instrument (PicoQuant, Germany). A 377 nm diode laser (PicoQuant, Germany) was used as the excitation source. The burst and normal modes embedded in the Time Harp 260 P module (PicoQuant, Germany) were employed for acquiring the signals from the Ir dopants and H host, respectively. The photoluminescence signals were obtained at 467 nm (Ir1 and Ir2), 460 nm (Ir3), 474 nm (Ir4), and 400 nm (H), through an automated motorized monochromator. Photoluminescence decay profiles were analyzed (OriginPro 2016, OriginLab) using a single exponential decay model.

**Electrochemical methods.** Cyclic, differential pulse, and second harmonic alternating current voltammetry experiments were carried out using a CH Instruments, CHI630 B instrument equipped with a three-electrode cell assembly. A Pt wire and a Pt microdisc were used as the counter and the working electrodes, respectively. An Ag/AgNO₃ couple was used as a pseudo reference electrode. Measurements were carried out in Ar-saturated THF (2.0 mL) using 0.10 M tetra-n-butylammonium hexafluorophosphate ($TBAPF_6$) as the supporting electrolyte at scan rates of 0.10 V s⁻¹ (cyclic voltammetry), 4.0 mV s⁻¹ (differential pulse voltammetry), and 25 mV s⁻¹ (second harmonic alternating current voltammetry). The concentration of the Ir dopants and H host was 2.0 mM. A ferrocenium/ferrocene couple was employed as the external reference. The potentials were reported as values against SCE, by adding 0.257 V. In the case of negative scans, validity of the observed peaks was examined by comparing voltammograms for a blank solution.

**Spectroelectrochemical measurements.** UV–vis–NIR absorption spectra of the radical species were obtained on an Agilent, Cary 5000 spectrophotometer with applying the anodic potentials (0.65 V vs. Ag/AgNO₃ for Ir1, 0.65 V vs. Ag/AgNO₃ for Ir2, 0.95 V vs. Ag/AgNO₃ for Ir3, and 0.80 V vs. Ag/AgNO₃ for Ir4), using the amperometric I–t curve method. A blank spectrum was taken for a 0.10 M $TBAPF_6$ solution (THF) in a spectroelectrochemical cell (path length = 0.5 mm) equipped with a Pt mesh working electrode, a Pt wire counter electrode, and an Ag/AnNO₃ pseudo reference electrode. 500 μL of a 2.0 mM Ir dopant solution was delivered into the spectroelectrochemical cell for the measurement.

**Stern–Volmer experiments.** Fluorescence quenching experiments were performed for a THF solution containing 100 μM H with added Ir (0–60 μM). Photoexcitation wavelength at 300 nm produced fluorescence emission of H, exclusively. Therefore, a monotonic decrease in the H emission was observed upon increasing the concentration of the Ir dopant. The decrease was quantitated as $I/I_0$, where $I$ and $I_0$ are the fluorescence emission of H in the presence and absence of the Ir dopant, respectively. The $I$ values were corrected by considering the absorbance of

**Fig. 5** Degradation of radical ion pairs. **a** UV–vis absorption difference spectra of a deaerated THF solution containing 3.0 mM H and 100 μM Ir4 during continuous photoirradiation using a 300 W Xenon lamp for 10 min. **b** UV–vis absorption difference spectra of a vacuum evaporated thin film of H molecularly doped with 15 wt % Ir4 during continuous photoirradiation using a 300 W Xenon lamp for 10 min. **c** UV–vis absorption difference spectra of a spincoated poly(methyl methacrylate) film molecularly doped with 15 wt % Ir4 during continuous photoirradiation using a 300 W Xenon lamp for 10 min. **d** Evolution of liquid chromatograms taken for the THF solution containing 3.0 mM H and 100 μM Ir4 during the continuous photoirradiation using a 300 W Xenon lamp for 10 min. **e** Comparison of the liquid chromatograms of the THF solution of 3.0 mM H and 100 μM Ir4 before (black) and after (magenta) the photoirradiation (10 min). The peaks marked with asterisks were byproducts and subjected to mass analyses. **f** Electrospray mass spectrum (positive mode) of the photolyzed THF solution containing 3.0 mM H and 100 μM Ir4. The inset structures correspond to the observed m/z values (a.m.u.). The inset graph shows a comparison of the isotopic distribution of the peak at m/z 827.39 (black bars) with theoretical values calculated for the structure shown on the left (magenta bars). **g** Electrospray mass spectrum (positive mode) of an oxidatively electrolyzed THF solution of 2.0 mM Ir4. The inset structures correspond to the observed m/z values (a.m.u.)

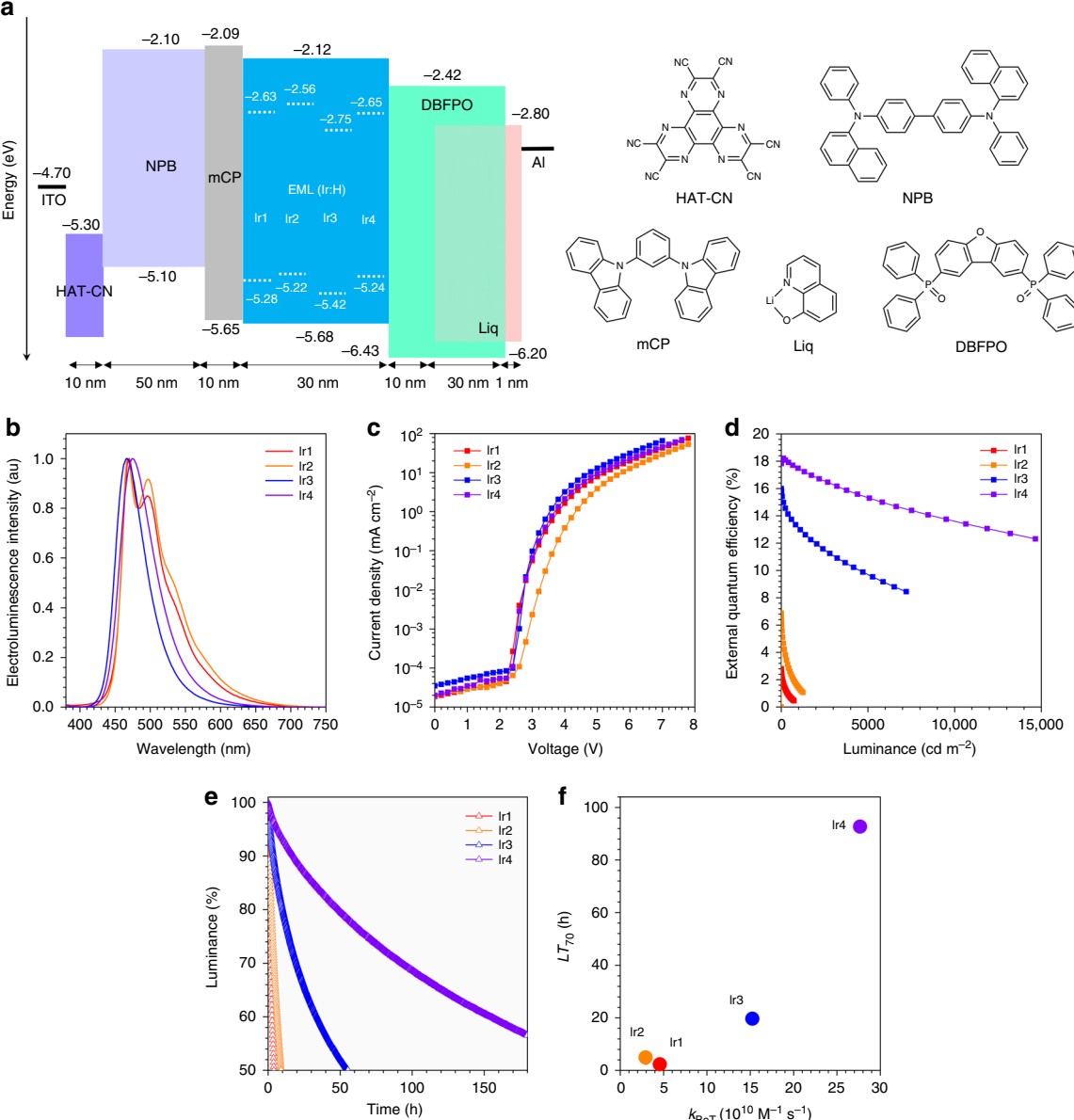

**Fig. 6** Device performance. **a** Schematic diagram of the configuration of the electroluminescence devices tested, including the energy levels of their component materials. **b** Electroluminescence spectra. **c** Current density–voltage curves. **d** External quantum efficiencies as a function of luminance. **e** Luminance decays during operation of the devices in a constant current driving mode. **f** Correlation between the operation lifetime ($LT_{70}$) of the devices and the rate constant for back electron transfer ($k_{BeT}$)

the added Ir dopant, following the relationship $I = I_{obs} \times (Abs/Abs_0) \times 1/(1 - 10^{-Abs})$ where $I_{obs}$, $Abs$, and $Abs_0$ are the observed fluorescence intensity, the absorbance of H at 300 nm in the presence of Ir, and the absorbance of H at 300 nm in the absence of Ir, respectively. The $I_0/I$ values were fit to the Stern–Volmer equation $I_0/I = (1 + K_a \cdot [Ir]) \times (1 + k_q \cdot \tau_0 \cdot [Ir])$. In this equation, $K_a$, $k_q$, $\tau_0$, and [Ir] are the association constant, the quenching constant, the fluorescence lifetime in the absence of Ir (3.8 ns), and the molar concentration of Ir, respectively.

**Calculation methods.** Geometry optimization was performed using Becke's three-parameter B3LYP exchange-correlation functional modified with the Coulomb-attenuated method (CAM–B3LYP), the double-ξ quality LANL2DZ basis set for the Ir atom, and the 6–311 + G(d, p) basis set for all the other atoms. A pseudo potential (LANL2DZ) was applied to replace the inner core electrons of the Ir atom, leaving the outer core $[(5 s)^2(5p)^6]$ electrons and the $(5d)^6$ valence electrons. Frequency calculations were subsequently performed to assess the stability of the convergence. Time-dependent density functional theory (TD–DFT) calculations were carried out for the optimized geometries using the same functional and basis sets. Geometry optimization and single-point calculations were performed using the Gaussian 09 program[39]. GaussSum was employed for simulation of the predicted electronic absorption spectra[40].

**Photoinduced EPR measurements.** A deaerated THF solution containing 1.0 mM compound was delivered into an EPR cell (i.d. = 0.7 mm). EPR spectra were recorded on a Bruker, EMX plus 6/1 spectrometer equipped with an Oxford Instrument, ESR 900 liquid He cryostat using an Oxford, ITC 513 temperature controller at 15 K under photoirradiation at a wavelength of ~300 nm (Korea Basic Science Institute, Western Seoul Center, Seoul).

**Laser flash photolysis.** An Ar-saturated THF solution in a quartz cell (path length = 1.0 cm) was excited by a Nd:YAG laser (EKSPLA, NT342) at a wavelength of 355 nm with 20 mJ pulse$^{-1}$. No positive transient signal was observed for dopant-only solutions under the measurement conditions. Time courses of the transient absorption were measured using Hamamatsu, photomultiplier tube R2949/InGaAs photodiode as detectors. The output from the detectors was recorded with a Tektronix, TDS3032 digitized oscilloscope. All experiments were performed at 298 K. The decays of the positive ΔAbs signals of Ir$^{\bullet+}$ were analyzed through a second-order kinetics model. Briefly, the molar absorbance ($\varepsilon$) of Ir$^{\bullet+}$ determined from the spectroelectrochemical measurement was divided by ΔAbs values at 1100 nm. The $\varepsilon$/ΔAbs data were plotted as a function of delay time, and fit to a linear line. The slope (in M$^{-1}$ s$^{-1}$) of the linear line corresponded to the $k_{BeT}$ value. The values were plotted as a function of $-\Delta G_{BeT}$, and correlated with

parabolic curves calculated from the Marcus equation for adiabatic outer-sphere electron transfer, $k_{BeT} = Z \exp[-(\Delta G_{BeT} + \lambda)^2 / 4\lambda k_B T]$, in the $-\Delta G_{BeT}$ range 0–3.0 eV. In this equation, $Z$, $k_B$, and $T$ are the collisional frequency taken as $1.0 \times 10^{12}$ M$^{-1}$ s$^{-1}$, the Boltzmann constant, and absolute temperature (298 K), respectively.

**Steady-state photolysis.** An Ar-saturated THF solution (3.0 mL) containing 3.0 mM H and 100 μM Ir was photoirradiated under a broad-band light from a Xenon lamp (300 W, Asahi Spectra, Max 303) for 10 min. A color change from yellow to orange was observed during the photolysis, which was monitored through steady-state UV–vis absorption spectroscopy. Photolysis was also performed for vacuum-evaporated films of H molecularly dispersed with 10 or 15 wt % Ir (glass substrate). The films were photoilluminated under 325 nm (3.5 mW, He-Cd laser, Kimmon Koha, IK3202R-D) or the broad-band light from the Xenon lamp, during which the changes in the photoluminescence (325 nm irradiated films) or UV–vis absorption (broad-band illuminated films) spectra were monitored. PMMA films molecularly doped with Ir dopants (15 wt %) were prepared and served as controls.

**Degradation product analyses.** HPLC experiments were performed on an Agilent, 6120 DW LC/MSD instrument equipped with a Poroshell, EC-C18 column. The photolyzed solutions were diluted in HPLC grade CH$_3$CN (1:9, v/v), and passed through a membrane filter (pore size = 8.0 μm) prior to injection. A 5 μL was injected and allowed to pass through the column at room temperature, using an eluent gradiently increased fractions of CH$_3$CN in H$_2$O. Chromatographic detection was performed with employing a UV detector ($\lambda_{obs}$ = 254 nm). Electrospray ionization mass analyses were subsequently performed at a positive ion detection mode (voltage = 70 V) in the range 200–1500 a.m.u. The photoirradiated films of H and Ir were dissolved in CH$_3$CN (HPLC grade) for the HPLC analyses.

**Device fabrication and characterization.** The organic layers used were deposited consecutively on pre-cleaned ITO glass substrates by employing a thermal evaporation system at a pressure $<1.0 \times 10^{-6}$ torr. A 1-nm-thick Liq layer and a 100-nm-thick Al layer were deposited as a cathode through thermal evaporation. The deposition rates of the organic and metal layers were 0.1 and 0.5 nm s$^{-1}$, respectively. Deposition of Liq was carried out at a rate 0.01 nm s$^{-1}$. The active device area of 4 mm$^2$ was defined by the area of an overlap between the ITO and Al electrodes. Current, voltage, and luminance of the devices were measured with a system consisting of a Keithley, 2400 Source-Mete,r and a PR-650 spectroradiometer. Operational lifetime measurements of the devices were taken in a constant current mode. $LT_{70}$ values were determined from the decay traces of % luminance plotted as a function of operation time. Operation time at which the % luminance decreased to 70% corresponded to $LT_{70}$.

**Data availability.** The data that support the findings of this study are available from the corresponding author upon reasonable request.

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

## Acknowledgements

This work was supported by the Samsung Advanced Institute of Technology, Samsung Electronics Co., Ltd.

## Author contributions

Si.K. designed and performed most of the experiments, analyzed the data, and wrote the manuscript. H.B., S.P., W.K., Y.J., and C.N. designed, synthesized, and characterized the materials. J.K. and J.S.K. fabricated and tested the devices. S.S. performed photophysical experiments. S.-G.I. coordinated the material preparation and OLED experiments. Su.K. supervised the work at Samsung Advanced Institute of Technology, and organized the project with Ewha Womans University. Y.Y. coordinated all of the experiments and analyses, and Y.Y. and S.-G.I. co-wrote the manuscript. All authors contributed to discussion on the study, and edited the manuscript.

## Additional information

**Competing interests:** The authors declare no competing interests.

