## [Peer Review File(PDF 1094 kb) · Nature Communications]

Reviewer #1 (Remarks to the Author):

The aim of this paper is to reveal the degradation mechanism of OLEDs. The analysis is very important from the viewpoints not only of fundamental science but also of the commercial applications, especially for blue-emitting OLEDs. Therefore, such kinds of studies are highly welcome. However, I find a crucial problem in this work; all the analyses were carried out in solutions, not in the devices. We cannot expect that phenomena in the solutions and in the devices are the same. The data in Fig. 4e show some correlation, but it has only four data points. Also, we do not know the correlation is causal relationship or not. I also would like to know why the authors select the iridium dopants in this manuscript. The PLQYs are not so high and the emission colors are not so deep. Additionally, the authors observed the C-N bond scission. It should be crucial for the device lifetime, but Fig. 4e indicates that LT95 can be explained without considering the bond scission. The analyses in solutions has some importance, so I recommend the publication in a more OLED focused journal.

Reviewer #2 (Remarks to the Author):

This manuscript by S. Kim et al addresses one of the longest standing questions in the OLED community; i.e. the root cause of the limited stability of blue phosphorescent OLEDs. Through a series of carefully conducted experiments they shed the light on a new (and very possible) degradation mechanism in which the excited host molecule is reductively quenched by electron transfer from the dopant molecule, thereby forming a radical ion pair. Instead of the “safe” annihilation of this radical ion pair through back electron transfer and electron-hole recombination to produce light from the dopant (first route), the host radical anion and dopant radical cation also have a finite probability to undergo chemical decomposition, primarily through the scission of the C-N bonds (second route), and thereby cause irreversible degradation. Data from photoluminescence measurements as a function of host and dopant concentrations give credible evidence to this quenching mechanism. Photo-induced EPR measurements are used to give direct verification of radical formation. The results show a correlation between OLED stability and the rate constant of the electron-hole recombination process, k_{bet} . Optical absorption, liquid chromatography and mass spectroscopy measurements are used to verify and identify the chemical decomposition byproducts produced by the second route.

The notion that an “intermolecular pathway for radical generation” may be the root cause of degradation in OLEDs, and especially blue PhOLEDs, has indeed not been investigated (or to the best of my knowledge even considered) before. Therefore, the work is quite novel. The presented experimental results give credible support to the argument, suggesting that this may indeed be an

important degradation mechanism. Given the importance of this specific issue to the OLED community, the findings will likely be of high interest and impact.

Certain points, outlined below, seem to have been overlooked or not adequately addressed however. In my view, addressing these points will be helpful for substantiating the conclusions and further strengthening the argument, and should therefore be done before accepting this work for publication in Nature Communications.

1) Although the data in figure 5(e) shows a good correlation between LT95 and k_{bet} , the fact that different current levels are used in aging these devices makes any stability comparisons difficult. For example, it is possible that the devices with Ir1 or Ir2 exhibit much lower LT95 values not because of their smaller k_{bet} but simply because they are driven at much higher currents. In order to avoid this ambiguity, comparing the PLQY (photoluminescence quantum yield) degradation rate among the four host:dopant systems (under photo-excitation) and showing that a correlation exists between a slower PLQY degradation and a larger k_{bet} should be a more fundamental verification of this phenomenon and the proposed degradation mechanism.

2) In the proposed mechanism, the presence of the dopant molecules is a prerequisite for the formation of the radical ion pair, and thus for this degradation mechanism to proceed. From this standpoint one would expect the degradation rate - under electrical and/or optical excitation - to be slower in case of un-doped host materials and to increase as the dopant concentration increases. This is however contrary to what is widely observed and reported in the literature where the use of dopants generally increases OLED stability. Addressing this point, perhaps through providing additional experimental data and/or suitable arguments, would be necessary.

Reviewer #3 (Remarks to the Author):

This paper describes a study of a very important problem and that is the short device lifetimes for blue phosphorescent OLEDs. These short lifetimes are limiting the wider spread adoption of OLEDs in a wide range of flat panel displays. Right now they dominate the mobile phone market (or soon will), but they are slow to move in to replace LCDs in flat screen monitors and televisions, due in large part to the short lifetimes of blue emitting OLEDs. The authors propose a mechanism for the degradation of these devices, but I am afraid that their experimental work falls far short of what is

needed to publish this paper. I recommend that the editors reject this paper and ask the authors to go back to the lab and do more work before resubmitting it. I have a number of reasons for making this recommendation:

- The authors have chosen a nontraditional host material for blue phosphorescent OLEDs. They chose a cyano-substituted mCBP derivative that has been reported for use in TADF based devices. mCBP has been used very successfully with close analogs to the emitters they use here. I suspect that the cyano group is influencing their studies and I would like to see a comparison of their with the cyano-mCBP to unsubstituted mCBP.
- In the SI they give the synthesis and characterization, but only list NMR and low resolution mass spec. There is no measure of purity. They need to include some measure of the purity of the material, such as CHN analysis. Were the compounds purified by sublimation prior to deposition? This is the common process, but it is not mentioned here.
- The paper and SI do not give enough information for one to evaluate the data they presented. There is mention of the procedure followed to make and test the OLED, there is no mention of whether the irradiation studies of films with a Xe arc lamp were done in the air or under an inert atmosphere. How did they determine the extinction coefficient of the charged species from spectroelectrochemical methods? They say they did it, but never explain how. This is a nontrivial thing to get right. Writing a communication does not mean that you can leave all of the details out. A reader needs to be able to read the paper and repeat the work. This is not possible here.
- The electrochemistry of their host material is irreversible in both oxidation and reduction. That being the case they do not know what the potentials are for either process. Without determining if the irreversibility is due to kinetic limitations or an EC mechanism of decomposition the potentials they measure are worthless. They need to show that it is an EC mechanism if they want to use the potentials they get from the CV measurements. Until they have done this the discussion of the driving force for electron transfer should be ignored/deleted.
- Why is their Stern-Volmer plot nonlinear? Is it really the effect of host-dopant aggregation? If so, they should show the corresponding Beers Law plot to show the same effect in absorbance. Is there any evidence for this aggregation effect other than the quenching studies? This seems like a very important point that they have glossed over.
- The transient absorbance data is very weak. The polaron peaks are very small and not readily discerned from the baseline.
- The authors have decided that their back electron transfer processes are in the Marcus inverted region and plot them as such. They fit them to two different reorganization energies, i.e. 1.6-1.7 for the first two dopants and 2.1-2.2 for the second two. Where did these values come from? Are they simply the values that give curves that contain the data points (one data point per curve)? Again, that is a weak enough argument that this should be best left out as well.
- The authors attribute the degradation to radical ion species, and use oxidative electrolysis to study this. Unfortunately, there are many examples of materials that are completely stable in a device and unstable to electrochemical oxidation or reduction. Alq3 is a good example. This has a very long lifetime as an electron transporter, but has totally irreversible oxidation and reductive

couples. The solvent in the electrochemical process leads to ligand dissociation that does not take place in a thin film. Again, these studies are inconclusive.

- The authors compare L95 values for the four devices. Their plot makes this comparison impossible to see and is not the best way to evaluate these anyway. The authors should look at L70 or L50. Look at the inset to Figure 4e, it is impossible to see the L95
- The oxidation potential for the four dopants fall in the order $3 > 4 > 1,2$, but the OLED lifetimes are $4 > 3 > 1,2$. Their model says that the faster the back electron transfer is the longer the lifetime. Based on their numbers the fastest back electron transfer rate should be from 4, but 3 gives a clearly longer lifetime.

Referee 1

Comments:

The aim of this paper is to reveal the degradation mechanism of OLEDs. The analysis is very important from the viewpoints not only of fundamental science but also of the commercial applications, especially for blue-emitting OLEDs. Therefore, such kinds of studies are highly welcome. However, I find a crucial problem in this work; all the analyses were carried out in solutions, not in the devices. We cannot expect that phenomena in the solutions and in the devices are the same. The data in Fig. 4e show some correlation, but it has only four data points. Also, we do not know the correlation is causal relationship or not. I also would like to know why the authors select the iridium dopants in this manuscript. The PLQYs are not so high and the emission colors are not so deep. Additionally, the authors observed the C-N bond scission. It should be crucial for the device lifetime, but Fig. 4e indicates that LT95 can be explained without considering the bond scission. The analyses in solutions has some importance, so I recommend the publication in a more OLED focused journal.

Response and Revisions: We accept the criticism that most of our mechanistic studies were performed in solutions. We actually attempted spectroscopic experiments for vacuum-deposited films, but small optical densities prevented us to perform mechanistic studies. Further increasing the thickness made the films hazy, limiting reliability of the measurements. Due to these reasons, we decided to focus on solution samples for mechanistic investigations. However, the coincidence between the degradation behaviors of the solutions and the vacuum-deposited films strongly indicates involvement of the same mechanism (Supplementary Fig. 16–19). Note that our study is to establish a new mechanism which is capable of explaining charge-neutral generation and annihilation of polaronic species. It should be noted that our mechanism provides a successful molecular explanation for the inherent instability of blue-emitting OLEDs. This mechanism is unprecedented, and its validity is fully supported by our chemical results. We would like to emphasize that our mechanism does not depend on the state of materials (i.e., solution vs film), because it is primarily governed by intrinsic properties of materials, such as electrochemical potentials. Therefore, the concern regarding the state of samples can be dismissed.

We believe that one of the key utilities of our study lies on providing direct spectroscopic evidence for radical ion pairs. These include transient absorption and photoluminescence spectroscopy results, photoinduced EPR data, spectroelectrochemistry, chemical analyses on degradation products, and quantum chemical calculation based on time-dependent density functional theory. These results coherently pointed to the key role of radical ion pairs in the degradation processes. Our device studies provided added evidence. The linearity between the device lifetimes (LT_{70}) and the rate constant for charge recombination ($k_{B\&T}$) shown in Fig. 4e can be explained only by our mechanism. The annihilation kinetics predicts varying extents of the net accumulation of labile radical ion pairs, which is consistent with the different device lifetime. This adherence strongly indicates that our mechanism is valid and applicable to films.

Regarding the comment about our choice of the blue-phosphorescent Ir dopants, we disagree with the reviewer. While the blue-phosphorescent Ir(III) complexes cannot outperform the state-of-the-art fluorophores with respect to PLQY and CIE coordinates, their ability to harness triplet excitons promises high device efficiencies. Recent studies have shown that blue-

phosphorescent Ir(III) complexes having *N*-heterocyclocarbenic (NHC) ligands are indeed very promising (For examples, *Inorg. Chem.* **2005**, *44*, 7992; *Angew. Chem., Int. Ed.* **2008**, *47*, 4542; *Inorg. Chem.* **2013**, *52*, 10756; *Inorg. Chem.* **2015**, *54*, 161; *Nat. Mater.* **2016**, *15*, 92. For reviews, *Adv. Mater.* **2012**, *24*, 3169; *Chem. Soc. Rev.* **2014**, *43*, 3551; *J. Mater. Chem. C* **2015**, *3*, 913; *Coord. Chem. Rev.* **2016**, *310*, 154). The high ligand-field strength and the absence of fluorine atom(s) in the NHC ligands are the key structural features that enable efficient and stable blue OLEDs. Validity of this structural control is widely accepted in the society of inorganic chemistry. Therefore, the comment of inadequateness of using the NHC-Ir(III) complexes is far from the current research. On the basis of this consideration, we believe that our use of the NHC-Ir(III) complex dopants is suitable for the mechanistic studies of blue electrophosphorescence.

Finally, according to the comment of bond scission, we newly performed kinetic studies. The rate constants (k_2) for the bimolecular reaction between the host (**H**) and **Ir3** or **Ir4** dopants have been determined. Specifically, we monitored the decay of the intraligand charge-transfer (ILCT) transition band (390 nm; $\epsilon = 39,000 \text{ M}^{-1} \text{ cm}^{-1}$ for **Ir3** and $25,000 \text{ M}^{-1} \text{ cm}^{-1}$ for **Ir4**) of the NHC ligands in **Ir3** and **Ir4** (500 μM) in the presence of 2.5 mM **H** upon continuous photoirradiation (white light) from a 300 W Xenon lamp, because this decay indicated cleavage of the C–N bond. Second-order kinetics analyses yielded the k_2 values. As included in Supplementary Fig. 18 (also attached below), **Ir3** possesses a k_2 value ($522 \text{ M}^{-1} \text{ s}^{-1}$) greater than that ($170 \text{ M}^{-1} \text{ s}^{-1}$) of **Ir4**. Note that this order is opposite to the order in k_{bet} . This inverse relationship is fully consistent with our mechanism, because faster charge recombination within radical ion pairs alleviates the chance for degradation. Attempts to obtain k_2 values for the pairs of **H** and **Ir1** or **Ir2** were unsuccessful due to significant overlaps between the ILCT bands of the **Ir1** and **Ir2** dopants with the **H** absorption. Nevertheless, photoluminescence decays of the films of **H:Ir** dopants provided additional supporting evidence. We compared the rate of photoluminescence decays of the films under continuous photoirradiation at 325 nm. This excitation wavelength could dominantly produce host exciton. As included in Supplementary Fig. 19 (also attached below), the decay rates are in the order of **Ir1** > **Ir2** > **Ir3** > **Ir4**. Note that this order is an inverse of the order of k_{bet} , a behavior consistent with our mechanism. Taken our new and original results together, we conclude that the radical ion pair is the reactive key intermediate that undergoes C–N bond cleavage, and that annihilation of radical ion pair is crucial for device longevity.

Supplementary Figure 18. Comparison of the rates of photolysis of the pairs of H and Ir3 or Ir4. **a, c,** UV-vis absorption difference spectra of deaerated THF solutions (3.0 mL) containing 2.5 mM **H** and 500 μM **Ir3** (**a**) or 500 μM **Ir4** (**c**) during continuous photoirradiation using a 300 W Xenon lamp for 10 s. **b, d,** Second-order kinetics analyses of the absorption decays at 390 nm (i.e., $\epsilon/\Delta\text{Abs}$) of the photolyzed solutions containing 2.5 mM **H** and 500 μM **Ir3** (**b**) or 500 μM **Ir4** (**d**). The 390 nm absorption bands correspond to the intraligand charge transfer (ILCT) transition of **Ir3** and **Ir4**. The slopes of the linear fits of the second-order plots correspond to rate constants for bimolecular reactions (k_2) between **H** and **Ir**. The k_2 values are $522 \text{ M}^{-1} \text{ s}^{-1}$ and $170 \text{ M}^{-1} \text{ s}^{-1}$ for **Ir3** and **Ir4**, respectively. Comparison of the k_2 values demonstrates superior stability of the pair of **H** and **Ir4**. The molar absorbance (ϵ) of **Ir3** and **Ir4** at 390 nm are $39000 \text{ M}^{-1} \text{ cm}^{-1}$ and $25000 \text{ M}^{-1} \text{ cm}^{-1}$, respectively. Note that **H** does not absorb the 390 nm light. Experiments for the pairs of **H** and **Ir1** or **Ir2** were unsuccessful due to significant spectral overlaps with the **H** absorption.

Supplementary Figure 19. Comparison of the photoluminescence decay rates of H films molecularly dispersed with Ir1–Ir4. **a–d**, Changes in the photoluminescence spectra of the H films doped with 10 wt % Ir1 (a), 10 wt % Ir2 (b), 10 wt % Ir3 (c), or 10 wt % Ir4 (d) during continuous photoillumination (325 nm, 3.5 mW, He-Cd laser). **e**, Plots of photoluminescence (PL, excitation wavelength = 325 nm) decay traces of the films as a function of photoirradiation times. **f**, A correlation between k_{BeT} and LT_{85} . Here, LT_{85} refers to the time when the normalized PL intensity decreases to 85% of the initial value. The films were encapsulated with glasses under an insert atmosphere to avoid potential degradation by molecular oxygen or moisture.

Referee 2

Comments:

This manuscript by S. Kim et al addresses one of the longest standing questions in the OLED community; i.e. the root cause of the limited stability of blue phosphorescent OLEDs. Through a series of carefully conducted experiments they shed the light on a new (and very possible) degradation mechanism in which the excited host molecule is reductively quenched by electron transfer from the dopant molecule, thereby forming a radical ion pair. Instead of the “ safe ” annihilation of this radical ion pair through back electron transfer and electron-hole recombination to produce light from the dopant (first route), the host radical anion and dopant radical cation also have a finite probability to undergo chemical decomposition, primarily through the scission of the C-N bonds (second route), and thereby cause irreversible degradation. Data from photoluminescence measurements as a function of host and dopant concentrations give credible evidence to this quenching mechanism.

Photo-induced EPR measurements are used to give direct verification of radical formation. The results show a correlation between OLED stability and the rate constant of the electron-hole recombination process, k_{bet} . Optical absorption, liquid chromatography and mass spectroscopy measurements are used to verify and identify the chemical decomposition byproducts produced by the second route.

The notion that an “ intermolecular pathway for radical generation ” may be the root cause of degradation in OLEDs, and especially blue PhOLEDs, has indeed not been investigated (or to the best of my knowledge even considered) before. Therefore, the work is quite novel. The presented experimental results give credible support to the argument, suggesting that this may indeed be an important degradation mechanism. Given the importance of this specific issue to the OLED community, the findings will likely be of high interest and impact.

Certain points, outlined below, seem to have been overlooked or not adequately addressed however. In my view, addressing these points will be helpful for substantiating the conclusions and further strengthening the argument, and should therefore be done before accepting this work for publication in Nature Communications.

Response: We deeply appreciate the reviewer for the positive comments and valuable suggestions. According to his or her comments, we revised our manuscript.

1. Although the data in figure 5(e) shows a good correlation between LT_{95} and k_{bet} , the fact that different current levels are used in aging these devices makes any stability comparisons difficult. For example, it is possible that the devices with Ir1 or Ir2 exhibit much lower LT_{95} values not because of their smaller k_{bet} but simply because they are driven at much higher currents. In order to avoid this ambiguity, comparing the PLQY (photoluminescence quantum yield) degradation rate among the four host:dopant systems (under photo-excitation) and showing that a correlation exists between a slower PLQY degradation and a larger k_{bet} should be a more fundamental verification of this phenomenon and the proposed degradation mechanism.

Response and Revisions: We appreciate for the helpful suggestion. According to this suggestion, we performed photobleaching experiments for vacuum-deposited films of the host

(**H**) and dopant (**Ir**). The films were encapsulated with glasses under an oxygen- and moisture-free atmosphere to avoid degradation by external species. Several attempts to measure PLQY were unsuccessful, due to significant scattering of lights and inability of our integrating sphere to accommodate the glass-encapsulated films. Instead, decays of the photoluminescence of the films (i.e., **H:Ir1-4**) were compared under continuous photoillumination (325 nm, 3.5 mW, He-Cd laser). The results are attached below, and are also included as Supplementary Fig. 19. The PL decay rates are in the order of **Ir1** > **Ir2** > **Ir3** > **Ir4**. It was found that k_{BET} correlated linearly with LT_{85} which was the time when the photoluminescence decreased to 85% of the initial value (Supplementary Fig. 19f). This result is also consistent with the photolysis results newly obtained for the solution of **H** and **Ir3** (bimolecular rate constant, $k_2 = 522 \text{ M}^{-1} \text{ s}^{-1}$) and the solution of **H** and **Ir4** ($k_2 = 170 \text{ M}^{-1} \text{ s}^{-1}$) (See Supplementary Fig. 18). Taken the results together, we can rule out the current effect (i.e., charge carrier densities) as a dominant origin for the degradation trend observed for the series of the **H:Ir** devices. We added the following discussion in our revised manuscript: "Photoluminescence stability of the films of **H:Ir** also followed this trend, indicating that the intermolecular electron-transfer reactions, not different current levels in devices, were responsible for device lifetime (Supplementary Fig. 19)."

Supplementary Figure 19. Comparison of the photoluminescence decay rates of H films molecularly dispersed with Ir1–Ir4. **a–d**, Changes in the photoluminescence spectra of the H films doped with 10 wt % Ir1 (a), 10 wt % Ir2 (b), 10 wt % Ir3 (c), or 10 wt % Ir4 (d) during continuous photoillumination (325 nm, 3.5 mW, He-Cd laser). **e**, Plots of photoluminescence (PL, excitation wavelength = 325 nm) decay traces of the films as a function of photoirradiation times. **f**, A correlation between k_{BeT} and LT_{85} . Here, LT_{85} refers to the time when the normalized PL intensity decreases to 85% of the initial value. The films were encapsulated with glasses under an insert atmosphere to avoid potential degradation by molecular oxygen or moisture.

2. In the proposed mechanism, the presence of the dopant molecules is a prerequisite for the formation of the radical ion pair, and thus for this degradation mechanism to proceed. From this standpoint one would expect the degradation rate - under electrical and/or optical excitation - to be slower in case of un-doped

host materials and to increase as the dopant concentration increases. This is however contrary to what is widely observed and reported in the literature where the use of dopants generally increases OLED stability. Addressing this point, perhaps through providing additional experimental data and/or suitable arguments, would be necessary.

Response and Revisions: This is an insightful comment. It seems reasonable that reducing the dopant concentration would retard the degradation of devices if the host–dopant pairs are degradation centers. However, this may not be the case because our degradation mechanism involves not the ground-state host–dopant pairs but the pairs of host exciton and a dopant. We have shown that host exciton mediates formation of the unstable radical ion pairs. A competing pathway available for this host exciton is an energy transfer to a dopant. At high dopant concentrations, the latter process (i.e., energy transfer to a dopant) may dominate to quickly consume the host exciton. It is expected that fractions of dopants located within the Förster radii of the host exciton will increase at larger doping concentrations. Consequently, the probability of host exciton-dopant pairs that can undergo electron transfer may decrease, which may enable longer device lifetime. In addition, increasing the doping concentration may initiate other degradation mechanisms. Devices containing low concentrations of dopants require high current densities to maintain photon flux as bright as those of high doping concentrations. The high current densities facilitate exciton–polaron quenching, another degradation pathway. On the contrary, high doping concentrations can suppress such degradation, because 1) the emitting layers experience more balanced charge carrier densities, and 2) the exciton profiles in the emitting layers become more homogenous. Both effects are beneficial for improved operational lifetime of blue OLEDs. Taken together, it is anticipated that high doping concentrations will result in long device lifetime. Our results actually adhere consistently to this explanation. As piled in Supplementary Table 2, longer operation lifetime was recorded for the devices having high doping concentrations in all the cases of **Ir1–Ir4**. Our new device results obtained employing a mCBP host in place of **H** also followed this trend (Supplementary Table 3; also attached below).

Supplementary Table 3. Electroluminescence data for the devices involving emitting layers of mCBP:Ir

Device	Doping concentration (wt %)	V_d^a (V)	λ_{EL}^b (nm)	Color coordinates (CIE _x , CIE _y)	EQE _{max} ^c (%)	EQE ^d (%)	LT_{70}^e (h)
mCBP: Ir1	10	7.24	467	(0.167, 0.216)	6.0	1.6	0.68
	20	6.65	468	(0.170, 0.259)	8.5	1.6	0.33
mCBP: Ir2	10	7.19	466	(0.177, 0.256)	8.0	3.1	1.04
	20	7.21	467	(0.179, 0.273)	9.5	3.0	0.43
mCBP: Ir3	10	4.98	463	(0.138, 0.141)	20.0	18.2	6.88
	20	4.37	463	(0.139, 0.152)	17.4	16.9	11.34

YOUNGMIN YOU
Professor

TEL: +82-2-3277-4275
E-mail: odds2@ewha.ac.kr

mCBP:Ir4	10	5.53	471	(0.142, 0.217)	15.9	18.0	17.50
	20	4.77	473	(0.145, 0.238)	16.9	16.9	44.66

^aDriving voltage at 500 cd m⁻². ^bPeak wavelength. ^cMaximum EQE. ^dEQE determined at 500 cd m⁻².
^eOperation time when the luminance decreases to 70% of its initial value. Constant current mode.

Referee 3

Comments:

This paper describes a study of a very important problem and that is the short device lifetimes for blue phosphorescent OLEDs. These short lifetimes are limiting the wider spread adoption of OLEDs in a wide range of flat panel displays. Right now they dominate the mobile phone market (or soon will), but they are slow to move in to replace LCDs in flat screen monitors and televisions, due in large part to the short lifetimes of blue emitting OLEDs. The authors propose a mechanism for the degradation of these devices, but I am afraid that their experimental work falls far short of what is needed to publish this paper. I recommend that the editors reject this paper and ask the authors to go back to the lab and do more work before resubmitting it. I have a number of reasons for making this recommendation:

Response and Revisions: We have addressed each of the comments provided by this reviewer. Our point-by-point responses are attached, as follows.

1. The authors have chosen a nontraditional host material for blue phosphorescent OLEDs. They chose a cyano-substituted mCBP derivative that has been reported for use in TADF based devices. mCBP has been used very successfully with close analogs to the emitters they use here. I suspect that the cyano group is influencing their studies and I would like to see a comparison of their with the cyano-mCBP to unsubstituted mCBP.

Response and Revisions: This comment is wrong. However, according to this comment, we newly prepared and evaluated control devices containing mCBP as a host material. The device performance of the mCBP-devices are attached below, and are also included as Supplementary Table 3 in our revised manuscript. We copy the device results of mCBP-CN (**H** in our manuscript) from Supplementary Table 2 for comparison.

Supplementary Table 2. Electroluminescence data for the devices involving emitting layers of H:Ir

Device	Doping concentration (wt %)	V_d^a (V)	λ_{EL}^b (nm)	Color coordinates (CIE _x , CIE _y)	EQE _{max} ^c (%)	EQE ^d (%)	LT_{70}^e (h)
H:Ir1	10	7.31	470	(0.186, 0.338)	2.0	0.5	0.92
	20	6.55	470	(0.187, 0.349)	2.8	0.7	1.88
H:Ir2	10	6.28	467	(0.196, 0.350)	4.1	1.3	1.89
	20	5.59	468	(0.200, 0.366)	6.9	2.6	4.11
H:Ir3	10	4.11	466	(0.139, 0.165)	14.3	11.6	12.27
	20	3.91	466	(0.140, 0.176)	16.0	13.9	19.70
H:Ir4	10	3.94	473	(0.146, 0.246)	17.9	17.1	60.19
	20	3.86	475	(0.149, 0.264)	18.2	17.8	93.05

^aDriving voltage at 500 cd m⁻². ^bPeak wavelength. ^cMaximum EQE. ^dEQE determined at 500 cd m⁻². ^eOperation time when the luminance decreases to 70% of its initial value. Constant current mode.

Supplementary Table 3. Electroluminescence data for the devices involving emitting layers of mCBP:Ir

Device	Doping concentration (wt %)	V_d^a (V)	λ_{EL}^b (nm)	Color coordinates (CIE _x , CIE _y)	EQE _{max} ^c (%)	EQE ^d (%)	LT_{70}^e (h)
mCBP:Ir1	10	7.24	467	(0.167, 0.216)	6.0	1.6	0.68
	20	6.65	468	(0.170, 0.259)	8.5	1.6	0.33
mCBP:Ir2	10	7.19	466	(0.177, 0.256)	8.0	3.1	1.04
	20	7.21	467	(0.179, 0.273)	9.5	3.0	0.43
mCBP:Ir3	10	4.98	463	(0.138, 0.141)	20.0	18.2	6.88
	20	4.37	463	(0.139, 0.152)	17.4	16.9	11.34
mCBP:Ir4	10	5.53	471	(0.142, 0.217)	15.9	18.0	17.50
	20	4.77	473	(0.145, 0.238)	16.9	16.9	44.66

^aDriving voltage at 500 cd m⁻². ^bPeak wavelength. ^cMaximum EQE. ^dEQE determined at 500 cd m⁻². ^eOperation time when the luminance decreases to 70% of its initial value. Constant current mode.

Comparisons of the two tables reveal longer device lifetime for the **H**-devices relative to mCBP-ones. LT_{70} for the **H**:20 wt % **Ir4** device was as long as 93.05 h, whereas the value decreased to 44.66 h for the mCBP:20 wt % **Ir4** device. Devices comprising other **Ir** dopants displayed identical trends (i.e., longer LT_{70} values for the **H**:**Ir** devices over the mCBP:**Ir** devices). This trend is consistent with our mechanism. The superior device longevity of **H** is explained by considering the net accumulation rates of radical ion pairs. As summarized below, the ground-state redox potentials of **H** and mCBP are similar: oxidation potentials (E_{ox} , vs SCE) = 1.50 V (**H**) and 1.48 V (mCBP); reduction potentials (E_{red} , vs SCE) = -1.82 V (**H**) and -1.79 V (mCBP). However, mCBP possesses a S_1 energy greater than **H**. This high excited-state energy locates the excited-state reduction potential (E_{red}^*) of mCBP more positive than that of **H** by 0.60 V (See Supplementary Table 4; also attached below). Accordingly, one-electron transfer from **Ir** dopants to mCBP becomes more thermodynamically favored. Since the forward electron transfer should be located in the Marcus-normal region of electron transfer, the exciton of mCBP is more rapidly quenched by electron transfer to form a radical ion pair (i.e., [mCBP^{•-}**Ir**^{•+}]). On the contrary, the rates for annihilation of the radical ion pair by charge recombination (i.e., [mCBP^{•-}**Ir**^{•+}] → mCBP + **Ir**) are predicted to be comparable to those for **H**, due to similar E_{red} values of the hosts. Indeed, the driving forces for annihilation ($-\Delta G_{\text{BeT}} = e \cdot [E_{\text{red}}(\text{host}) - E_{\text{ox}}(\text{Ir})]$) are calculated to be 2.60–2.91 eV for the mCBP:**Ir** pairs, which are very similar to the values (2.63–2.94 eV) for the **H**:**Ir** pairs. As a result, radical ion species are more accumulated for the pairs of mCBP and **Ir**, compared to the pairs of **H** and **Ir**. These considerations predict shorter device lifetime of the mCBP-devices. This prediction is indeed in good accordance with the device results compiled in the above tables.

Supplementary Table 4. Electrochemical potentials of H and mCBP

Host	λ_{em} (nm)	E_{ox} (V vs SCE)	E_{red} (V vs SCE)	E_{red}^* (V vs SCE)
H	411	1.50	-1.82	1.28
mCBP	346	1.48	-1.79	1.88

Taken the results together, our new results reveal a benefit of the cyano group. The bandgap energy decreases due to the presence of the cyano group in our host. This electronic control produces a cathodic shift in the excited-state reduction potential (E_{red}^*) of **H** relative to that of mCBP, which decreases the driving force for formation of unstable radical ion pairs. Since the driving force for annihilation is similar, the electrochemical control enables less accumulation of radical ion pairs for **H**. We added the new results in the Supplementary Information (Supplementary Tables 3 and 4) and following discussion in our revised manuscript: “Finally, validity of our mechanism was further examined by comparing the LT_{70} values with those of devices having mCBP (3,3'-bis(9-carbazolyl)biphenyl) in place of **H** as a host material. It is predicted that radical ion pairs are more accumulated in the mCBP:**Ir** layers than the **H**:**Ir** layers, due to the more positive E_{red}^* (1.88 V vs SCE; c.f., E_{red}^* of **H** = 1.28 V vs SCE) and similar E_{red} (-1.79 V vs SCE; c.f., E_{red} of **H** = -1.82 V vs SCE) values of mCBP. Indeed, devices of mCBP exhibited poor longevity consistently for all the **Ir** dopants, as quantitated from shorter LT_{70} values (Supplementary Table 3).”

Finally, we would like to mention that we varied the structures of the dopants, not that of the host. Therefore, the conclusions obtained from our results should originate from the structural control of the dopants.

2. In the Si they give the synthesis and characterization, but only list NMR and low resolution mass spec. There is no measure of purity. They need to include some measure of the purity of the material, such as CHN analysis. Were the compounds purified by sublimation prior to deposition? This is the common process, but it is not mentioned here.

Response and Revisions: According to this comment, we examined the purity of the **Ir** dopants. High performance liquid chromatography experiments have been performed for all the synthesized **Ir** dopants. The analyses revealed high purity of our **Ir** dopants: purity of **Ir1** > 99.95%; purity of **Ir2** > 99.79%; purity of **Ir3** > 99.65%; purity of **Ir4** > 99.85%. The chromatograms are attached below, and are also included in our revised Supplementary Information (Supplementary Fig. 3, 6, 9, and 12). Finally, we also performed elemental (C, H, and N) analyses for the **Ir** dopants. The results revealed satisfactory purity of the compounds. The new elemental analyses have been included in our revised Supplementary Information, and are also attached below.

YOUNGMIN YOU
 Professor

TEL: +82-2-3277-4275
 E-mail: odds2@ewha.ac.kr

Supplementary Figure 3. High performance liquid chromatogram for Ir1. Purity was determined to be greater than 99.95%.

Supplementary Figure 6. High performance liquid chromatogram for Ir2. Purity was determined to be greater than 99.79%.

Supplementary Figure 9. High performance liquid chromatogram for Ir3. Purity was determined to be greater than 99.65%.

Supplementary Figure 12. High performance liquid chromatogram for Ir4. Purity was determined greater than 99.85%.

Summary of elemental analysis results:

- Ir1:** Anal. Calcd for C₆₀H₄₅IrN₆: C, 69.14; H, 4.35; N, 8.06. Found: C, 69.16; H, 4.34; N, 8.06.
- Ir2:** Anal. Calcd for C₆₀H₄₅IrN₆: C, 69.14; H, 4.35; N, 8.06. Found: C, 68.78; H, 4.35; N, 7.99.
- Ir3:** Anal. Calcd for C₅₇H₄₅IrN₁₂: C, 62.79; H, 4.16; N, 15.42. Found: C, 62.82; H, 4.21; N, 15.51.
- Ir4:** Anal. Calcd for C₈₁H₉₃IrN₁₂: C, 68.18; H, 6.57; N, 11.78. Found: C, 68.25; H, 6.59; N, 11.81.

3. The paper and Si do not give enough information for one to evaluate the data they presented. There is mention of the procedure followed to make and test the OLED, there is no mention of whether the

irradiation studies of films with a Xe arc lamp were done in the air or under an inert atmosphere. How did they determine the extinction coefficient of the charged species from spectroelectrochemical methods? They say they did it, but never explain how. This is a nontrivial thing to get right. Writing a communication does not mean that you can leave all of the details out. A reader needs to be able to read the paper and repeat the work. This is not possible here.

Response and Revisions: According to this suggestion, we thoroughly revised the “Methods” section in the main text. The revised Methods is also attached below:

Methods

Materials. Synthetic details for the **Ir** dopants are described in the Supplementary Section 1. **H** was prepared following the method reported previously.³⁷ Organic materials for device fabrication were purchased from commercial suppliers, and purified by sublimation at 10^{-5} torr prior to deposition. Films were deposited under high vacuum ($< 1.0 \times 10^{-6}$ torr) onto a pre-cleaned glass substrate (1 cm \times 1 cm). Poly(methyl methacrylate) (PMMA, Mw \sim 120,000, Sigma–Aldrich) films doped with the **Ir** dopants (15 wt %) were dissolved in 1,2-dichloroethane (5 wt % total solute relative to solution). The solution was sonicated for 30 min, and passed through a membrane filter (pore size = 8.0 μ m). An aliquot of the polymer solution was placed onto a pre-cleaned glass substrate, and was spin-casted with employing an EPLEX, SPIN-1200D spin coater. Spectrophotometric grade THF stored under an insert atmosphere was used for spectroscopic and electrochemical measurements.

Steady-state UV–vis absorption measurements. UV–vis absorption spectra were collected on an Agilent Cary 300 spectrophotometer. Solution samples were prepared in THF to a 10 μ M concentration prior to the measurements, unless otherwise stated.

Steady-state photoluminescence measurements. Photoluminescence spectra were obtained using a PTI, Quanta Master 400 scanning spectrofluorimeter at 298 K. The 10 μ M solutions or the films were used for the measurements. The **Ir** solutions were deaerated by bubbling Ar for > 15 min. Photoexcitation wavelengths were 341 nm (**Ir1**), 340 nm (**Ir2**), 396 nm (**Ir3**), 400 nm (**Ir4**), and 340 nm (**H**). The photoluminescence quantum yields (PLQYs) were determined employing an absolute PLQY measurement system (Hamamatsu, C11347-01).

Photoluminescence lifetime (τ_{obs}) measurements. Ar-saturated 50 μ M solutions in THF were used for the determination of the τ_{obs} values of the **Ir** dopants and **H** host. Photoluminescence decay traces were acquired based on time-correlated single-photon-counting (TCSPC) techniques using a FluoTime 200 instrument (PicoQuant, Germany). A 377 nm diode laser (PicoQuant, Germany) was used as the excitation source. The burst and normal modes embedded in the Time Harp 260P module (PicoQuant, Germany) were employed for acquiring the signals from the **Ir** dopants and **H** host, respectively. The photoluminescence signals were obtained at 467 nm (**Ir1** and **Ir2**), 460 nm (**Ir3**), 474 nm (**Ir4**), and 400 nm (**H**), through an automated motorized monochromator. Photoluminescence decay profiles were analyzed (OriginPro 2016, OriginLab) using a single exponential decay model.

Electrochemical methods. Cyclic, differential pulse, and second harmonic alternating current voltammetry experiments were carried out using a CHI630 B instrument (CH Instruments, Inc.) equipped with a three-electrode cell assembly. A Pt wire and a Pt microdisc were used as the counter and the working electrodes, respectively. An Ag/AgNO₃ couple was used as a pseudo reference electrode. Measurements were carried out in Ar-saturated THF (2.0 mL) using 0.10 M tetra-*n*-butylammonium hexafluorophosphate (TBAPF₆) as the supporting electrolyte at scan rates of 0.10 V s⁻¹ (cyclic voltammetry), 4.0 mV s⁻¹ (differential pulse voltammetry), and 25 mV s⁻¹ (second harmonic alternating current voltammetry). The concentration of the **Ir** dopants and **H** host was 2.0 mM. A ferrocenium/ferrocene couple was employed as the external reference. The potentials were reported as values against standard calomel electrode (SCE), by adding 0.257 V. In the case of negative scans, validity of the observed peaks was examined by comparing voltammograms for a blank solution.

Spectroelectrochemical measurements. UV-vis-NIR absorption spectra of the radical species were obtained on an Agilent, Cary 5000 spectrophotometer with applying the anodic potentials (0.65 V vs Ag/AgNO₃ for **Ir1**, 0.65 V vs Ag/AgNO₃ for **Ir2**, 0.95 V vs Ag/AgNO₃ for **Ir3**, and 0.80 V vs Ag/AgNO₃ for **Ir4**), using the amperometric *I*-*t* curve method. A blank spectrum was taken for a 0.10 M TBAPF₆ solution (THF) in a spectroelectrochemical cell (path length = 0.5 mm) equipped with a Pt mesh working electrode, a Pt wire counter electrode, and an Ag/AgNO₃ pseudo reference electrode. 500 μL of a 2.0 mM **Ir** dopant solution was delivered into the spectroelectrochemical cell for the measurement.

Stern-Volmer experiments. Fluorescence quenching experiments were performed for a THF solution containing 100 μM **H** with added **Ir** (0–60 μM). Photoexcitation wavelength at 300 nm produced fluorescence emission of **H**, exclusively. Therefore, a monotonic decrease in the **H** emission was observed upon increasing the concentration of the **Ir** dopant. The decrease was quantitated as *I*/*I*₀, where *I* and *I*₀ are the fluorescence emission of **H** in the presence and absence of the **Ir** dopant, respectively. The *I* values were corrected by considering the absorbance of the added **Ir** dopant, following the relationship $I = I_{\text{obs}} \times (Abs/Abs_0) \times 1/(1 - 10^{-Abs})$ where *I*_{obs}, *Abs*, and *Abs*₀ are the observed fluorescence intensity, the absorbance of **H** at 300 nm in the presence of **Ir**, and the absorbance of **H** at 300 nm in the absence of **Ir**, respectively. The *I*₀/*I* values were fit to the Stern-Volmer equation $I_0/I = (1 + K_a \cdot [Ir]) \times (1 + k_q \cdot \tau_0 \cdot [Ir])$. In this equation, *K*_a, *k*_q, *τ*₀, and [**Ir**] are the association constant, the quenching constant, the fluorescence lifetime in the absence of **Ir** (3.8 ns), and the molar concentration of **Ir**, respectively.

Calculation methods. Geometry optimization was performed using Becke's three-parameter B3LYP exchange-correlation functional modified with the Coulomb-attenuated method (CAM-B3LYP), the "double-ξ" quality LANL2DZ basis set for the Ir atom, and the 6-311+G(d,p) basis set for all the other atoms. A pseudo potential (LANL2DZ) was applied to replace the inner core electrons of the Ir atom, leaving the outer core [(5s)²(5p)⁶] electrons and the (5d)⁶ valence electrons. An *N,N*-trans structure was employed as the starting geometry of the **Ir** dopant. Frequency calculations were subsequently performed to assess the stability of the convergence. Time-dependent density functional theory (TD-DFT) calculations were carried out for the optimized geometries using the same functional and basis sets. Geometry optimization and single-point calculations were performed using the Gaussian 09 program.³⁹ GaussSum was employed for simulation of the predicted electronic absorption spectra.⁴⁰

Photoinduced electron paramagnetic resonance (EPR) measurements. A deaerated THF solution containing 1.0 mM compound was delivered into an EPR cell (i.d. = 0.7 mm). EPR spectra were recorded on a Bruker, EMX plus 6/1 spectrometer equipped with an Oxford Instrument, ESR 900 liquid He cryostat using an Oxford ITC 513 temperature controller at 15 K under photoirradiation at a wavelength of ~ 300 nm (Korea Basic Science Institute, Western Seoul Center, Seoul).

Laser flash photolysis. An Ar-saturated THF solution in a quartz cell (path length = 1.0 cm) was excited by a Nd:YAG laser (EKSPLA, NT342) at a wavelength of 355 nm with 20 mJ/pulse. No positive transient signal was observed for dopant-only solutions under the measurement conditions. Time courses of the transient absorption were measured using Hamamatsu, photomultiplier tube R2949/InGaAs photodiode as detectors. The output from the detectors was recorded with a Tektronix, TDS3032 digitized oscilloscope. All experiments were performed at 298 K. The decays of the positive ΔAbs signals of $\text{Ir}^{\bullet+}$ were analyzed through a second-order kinetics model. Briefly, the molar absorbance (ϵ) of $\text{Ir}^{\bullet+}$ determined from the spectroelectrochemical measurement was divided by ΔAbs values at 1100 nm. The $\epsilon/\Delta\text{Abs}$ data were plotted as a function of delay time, and fit to a linear line. The slope (in $\text{M}^{-1} \text{s}^{-1}$) of the linear line corresponded to the k_{BeT} value. The values were plotted as a function of $-\Delta G_{\text{BeT}}$, and correlated with parabolic curves calculated from the Marcus equation for adiabatic outer-sphere electron transfer, $k_{\text{BeT}} = Z \exp[-(\Delta G_{\text{BeT}} + \lambda)^2/4\lambda k_{\text{B}}T]$, in the $-\Delta G_{\text{BeT}}$ range 0–3.0 eV. In this equation, Z , k_{B} , and T are the collisional frequency taken as $1.0 \times 10^{12} \text{ M}^{-1} \text{ s}^{-1}$, the Boltzmann constant, and absolute temperature (298 K), respectively.

Steady-state photolysis. An Ar-saturated THF solution (3.0 mL) containing 3.0 mM **H** and 100 μM **Ir** was photoirradiated under a broad-band light from a Xenon lamp (300 W, Asahi Spectra, Max 303) for 10 min. A color change from yellow to orange was observed during the photolysis, which was monitored through steady-state UV–vis absorption spectroscopy. Photolysis was also performed for vacuum-evaporated films of **H** molecularly dispersed with 10 or 15 wt % **Ir** (glass substrate). The films were photoilluminated under 325 nm (3.5 mW, He-Cd laser, Kimmon Koha, IK3202R-D) or the broad-band light, during which the changes in the photoluminescence (325 nm irradiated films) or UV–vis absorption (broad-band illuminated films) spectra were monitored. PMMA films molecularly doped with **Ir** dopants (15 wt %) were prepared and served as controls.

Degradation product analyses. High performance liquid chromatography (HPLC) experiments were performed on an Agilent, 6120 DW LC/MSD instrument equipped with a Poroshell, EC-C18 column. The photolyzed solutions were diluted in HPLC grade CH_3CN (1:9, v/v), and passed through a membrane filter (pore size = $8.0 \mu\text{m}$) prior to injection. A $5 \mu\text{L}$ was injected and allowed to pass through the column at room temperature, using an eluent gradually increased fractions of CH_3CN in H_2O . Chromatographic detection was performed with employing a UV detector ($\lambda_{\text{obs}} = 254 \text{ nm}$). Electrospray ionization mass analyses were subsequently performed at a positive ion detection mode (voltage = 70 V) in the range 200–1500 amu. The photoirradiated films of **H** and **Ir** were dissolved in CH_3CN (HPLC grade) for the HPLC analyses.

Device fabrication and characterization. The organic layers used were deposited consecutively on pre-cleaned ITO glass substrates by employing a thermal evaporation system at a pressure $< 1.0 \times 10^{-6}$ torr. A 1 nm-thick Liq layer and a 100 nm-thick Al layer were deposited as a cathode through thermal evaporation. The deposition rates of the organic and metal layers were 0.1 nm s^{-1} and 0.5 nm s^{-1} , respectively. Deposition of Liq was carried out at a rate 0.01 nm s^{-1} . The active device area of 4 mm^2 was defined by the area of an overlap between the ITO and Al electrodes. Current, voltage, and luminance of the devices were measured with a system consisting of a Keithley, 2400 Source-Meter and a PR-650 spectroradiometer. Operational lifetime measurements of the devices were taken in a constant current mode. LT_{70} values were determined from the decay traces of % luminance plotted as a function of operation time. Operation time at which the % luminance decreased to 70% corresponded to LT_{70} .

4. The electrochemistry of their host material is irreversible in both oxidation and reduction. That being the case they do not know what the potentials are for either process. Without determining if the irreversibility is due to kinetic limitations or an EC mechanism of decomposition the potentials they measure are worthless. They need to show that it is an EC mechanism if they want to use the potentials they get from the CV measurements. Until they have done this the discussion of the driving force for electron transfer should be ignored/deleted.

Response and Revisions: As we indicated in our manuscript, the host material displayed irreversible processes in both oxidation and reduction. We performed differential pulse voltammetry (DPV) experiments, and reported the values in our original manuscript. Validity of the potential values was also examined by performing second harmonic alternating current voltammetry (SHACV), and the result was consistent with our original potential value (see the voltammograms attached below). Validity of the potential values is also demonstrated from the similarity of the optical bandgap energy (3.1 eV) and the difference between the oxidation and reduction potentials (3.32 V) of H. Taken the results together, we conclude that the potential values are reliable. The SHACV results, together with the DPV voltammogram, are included in Supplementary Fig. 13.

Supplementary Figure 13. Determination of the reduction potential of H. Differential pulse voltammogram (**a**, scan rate = 4.0 mV s^{-1}) and second harmonic alternating current voltammogram (**b**, scan rate = 25 mV s^{-1} and frequency = 100 Hz) of 2.0 mM H dissolved in an Ar-saturated THF solution (2.0 mL) containing 0.10 M TBAPF_6 electrolyte. The **H** solution was delivered to a standard three-electrode cell assembly equipped with a Pt wire counter electrode, a Pt disc working electrode, and an Ag/AgNO₃ pseudo reference electrode. The dotted lines in **a** and **b** are the voltammograms of blank solutions (i.e., no **H**).

5. Why is their Stern-Volmer plot nonlinear? Is there really the effect of host-dopant aggregation? If so, they should show the corresponding Beers Law plot to show the same effect in absorbance. Is there any evidence for this aggregation effect other than the quenching studies? This seems like a very important point that they have glossed over.

Response and Revisions: The association between the photoexcited host and the dopant may be inferred from the non-linear behavior in the Stern–Volmer plot. As pointed out by this reviewer, pre-association may be important because it facilitates electron transfer. According to this comment, we measured UV–vis absorption spectra of equimolar solutions of the host (**H**) and dopant (**Ir3**) with increasing the concentration of **Ir3** ($0, 20, 50, 100, 250, 500, 1000,$ and $2000\text{ }\mu\text{M}$). The results are attached below. As seen from the results, no discernable change was found in the UV–vis absorption spectra. The absorption spectra remained as a mathematical sum of individual spectra of **H** and **Ir3**. Any absorption bands due to the formation of aggregates and charge-transfer complexes were not observed. The invariability indicates the absence of ground-state interactions, and may originate from the facts that 1) the octahedral Ir(III) complexes hardly form intermolecular arrangements for π – π interactions, and 2) **H** and **Ir3** cannot make charge-transfer interactions due to the unfavorable potential alignment. Actually, the electrochemical potentials (Table 1) predict thermodynamic disallowance for ‘ground-state’ charge-transfer interactions. Taken together, it would be concluded that the association between **H** and **Ir3** occurs exclusively in the excited state of **H**. The non-linear behavior on the

Stern–Volmer plot may be due to different distance effects of energy and electron transfer in this excited-state encounter complex with Ir.

Figure. Examination of the occurrence of ground-state interactions between H and Ir3. **a**, UV–vis absorption spectra of THF solutions (3.0 mL) containing equimolar mixtures of H and Ir3. The concentration of H (or Ir3) was increased from 20 μM to 2.0 mM to probe the existence of any ground-state interactions. The UV–vis absorption spectra of H (dashed line) and Ir3 (dotted line) are also included for comparison. The absorption signals were saturated at high concentrations (encircled in red). **b**, A magnified view of the spectra in Fig. a. The rises of the red (20 μM H + 20 μM Ir3) and orange (50 μM H + 50 μM Ir3) curves in the visible region was due to instrumental noises.

Finally, we included the Stern–Volmer analysis data for the pairs of H and other Ir dopants in our revised manuscript. The results are shown in Supplementary Fig. 14. We also revised the caption of Fig. 2 to involve the fit parameters, as follows: “The I_0/I values were fit to the Stern–Volmer equation $I_0/I = (1 + K_a \cdot [\text{Ir3}]) \times (1 + k_q \cdot \tau_0 \cdot [\text{Ir3}])$. In this equation, K_a , k_q , τ_0 , and $[\text{Ir3}]$ are the association constant, the quenching constant, the fluorescence lifetime in the absence of Ir3 (3.8 ns), and the molar concentration of Ir3, respectively. The Stern–Volmer analysis yielded k_q and K_a to be $> 10^{12} \text{ M}^{-1} \text{ s}^{-1}$ and $6.8 \times 10^3 \text{ M}^{-1}$, respectively. The non-negligible contribution of the static quenching (i.e., the $(1 + K_a \cdot [\text{Ir3}])$ term in the Stern–Volmer equation) may indicate the existence of excited-state interactions between H and Ir3. Stern–Volmer analyses for other Ir dopants are shown in Supplementary Fig. 14.”

6. The transient absorbance data is very weak. The polaron peaks are very small and not readily discerned from the baseline.

Response: As indicate by the reviewer, the transient absorption spectra were weak. Our attempt to improve the signal-to-noise ratio by increasing the laser power was unsuccessful, as quartz cells started to be damaged. This weak visibility originated from the presence of two fast

processes competing with excited-state electron transfer. The processes included intramolecular relaxation of the host exciton and energy transfer to dopant. Although we did not determine the quantum yield for the generation of radical ion pairs, the value must remain small due to these competing processes. Actually, the **H:Ir** system retained high luminescence both in solutions and devices, which indicated small populations of radical ion pairs. Despite the small intensities, our spectral assignments of the transient absorption signals are fully supported by a variety of evidence. Fig. 3b includes the UV-vis-NIR spectroelectrochemical result (solid curve) and simulated electronic transition spectra (blue and red curves) calculated based on time-dependent density functional theory. These results coherently pointed to the notion that the transient absorption spectra were due to the radical cation of the **Ir** dopants. Therefore, we are confident of validity of our transient absorption spectra.

7. The authors have decided that their back electron transfer processes are in the Marcus inverted region and plot them as such. They fit them to two different reorganization energies, i.e. 1.6-1.7 for the first two dopants and 2.1-2.2 for the second two. Where did these values come from? Are they simply the values that give curves that contain the data points (one data point per curve)? Again, that is a weak enough argument that this should be best be left out as well.

Response and Revisions: As we already indicated in our manuscript, we did not numerically fit the data point to the Marcus equation for outer-sphere, adiabatic bimolecular electron transfer. The rates of back electron transfer were distributed over the two different regions, depending on the structures of the NHC ligands. This scattered and small population of data points prohibited us to obtain numerical fits to the Marcus equation. The solid parabolas shown in Fig. 3d are calculated curves of the Marcus equation in the region $-\Delta G_{\text{BeT}} = 0-3.0$ eV. They were shown for visual guidance for rough estimation of the reorganization energies. We admit that our initial explanation may mislead the readers. To avoid misunderstanding, we revised the caption of Fig. 3d as follows: “**d**, Plot of the rate constant for back electron transfer for charge recombination (k_{BeT} , black squares) as a function of the driving force and the Marcus plots for outer-sphere electron transfer calculated for the reorganization energies (λ) at 1.7 eV (red), 1.9 eV (orange), 2.2 eV (green), and 2.3 eV (blue). The Marcus plots were constructed from the equation, $k_{\text{BeT}} = Z \exp[-(\Delta G_{\text{BeT}} + \lambda)^2/4\lambda k_{\text{B}}T]$. In this equation, Z , k_{B} , and T are the collisional frequency taken as $1.0 \times 10^{12} \text{ M}^{-1} \text{ s}^{-1}$, the Boltzmann constant, and absolute temperature (298 K), respectively.”

8. The authors attribute the degradation to radical ion species, and use oxidative electrolysis to study this. Unfortunately, there are many examples of materials that are completely stable in a device and unstable to electrochemical oxidation or reduction. Alq3 is a good example. This has a very long lifetime as an electron transporter, but has totally irreversible oxidation and reductive couples. The solvent in the electrochemical process leads to ligand dissociation that does not take place in a thin film. Again, these studies are inconclusive.

Response: This is misunderstanding of our mechanism. Solvent is not directly involved in any of the degradation processes in our mechanism. Exciton-mediated electron transfer between the

host and dopant molecules produces radical ion pair, the key degradation intermediate. Our results unambiguously indicated that the degradation was mediated by this radical ion pair (Supplementary Fig. 15–19). Therefore, intermolecular radical ion pairs, not solvent, are directly involved in the degradation.

It may be true that electrochemical stability of a material is not necessarily linked to device stability. However, this cannot rebut the claim of the reviewer because in many cases materials are placed in devices at controlled space charge conditions. Alq₃ is a good example. As mentioned by the reviewer, Alq₃ serves as a very stable electron-transporting material. This stability is due to the facts that i) Alq₃ layers are not located within hole-rich regions, so decomposition from the radical cation of Alq₃ is suppressed, and ii) Alq₃ exhibits electron mobility two orders of magnitude greater than hole mobility, so build-up of electron space charge within the Alq₃ layer is kinetically avoided. These two conditions effectively minimize the generation of radical ion species of Alq₃ during device operation. In fact, one can find papers which linked instability of Alq₃ radical cation with electrochemical irreversibility (For example, see *Science*, **1999**, 283, 1900; *Chem. Mater.* **1996**, 8, 1363; *J. Appl. Phys.* **2003**, 93, 1108; *J. Appl. Phys.* **2001**, 89, 4673). It is notable that Poppvic and co-workers improved device stability by minimizing the density of hole carriers within an Alq₃ layer (*Science*, **1999**, 283, 1900). The recent study by Aziz also revealed that Alq₃ underwent degradation upon exposure to electron space charge (*Chem. Mater.* **2007**, 19, 2079). In particular, this study demonstrated that polaronic states of Alq₃ were vulnerable to irreversible degradation, regardless of the identity of charge (i.e., radical cation or radical anion). These considerations force us to avoid ruling out electrochemical stability as the key requirement of materials for device stability.

In fact, radical ion species have been frequently identified as the key intermediates for intrinsic degradation of several emitters and charge-transporting materials. There are numerous recent experimental studies which indicated that radical ion species are vulnerable to bond dissociation. The groups of Aziz and Forrest pioneered mechanistic studies of polaronic degradation. For examples, see *Appl. Phys. Lett.* **2010**, 97, 063309; *Org. Electron.* **2012**, 13, 2075; *Adv. Funct. Mater.* **2013**, 23, 2239; *J. Appl. Phys.* **2012**, 112, 064502; *Appl. Phys. Lett.* **2012**, 101, 173502; *Org. Electron.* **2011**, 12, 2056; *J. Appl. Phys.* **2011**, 109, 044501; *Org. Electron.* **2011**, 12, 2056; *J. Appl. Phys.* **2011**, 109, 044501; *Appl. Phys. Lett.* **2010**, 97, 063309; *J. Appl. Phys.* **2009**, 105, 124514. Tang and co-workers also suggested that FIrpic, a representative blue-phosphorescent dopant, became unstable upon one-electron oxidation (*Org. Electron.* **2014**, 15, 1312). Note that FIrpic displays an irreversible oxidation process. In addition, Kodankov and co-worker claimed that degradation of the NPB hole-transporting layer was due to the formation of interlayer radical ion species (*J. Appl. Phys.* **2010**, 108, 074513). These studies provide firm evidence that radical species mediate intrinsic degradation of materials. Such understanding recently inspired the Li and co-workers to achieve superior stability of phosphorescent OLEDs (*Adv. Mater.* **2016**, 29, 1605002).

Finally, we would like to comment several recent progresses on quantum chemical calculations on the stability of materials in OLEDs. The groups of Brédas and Van Vooris reported in their independent research that bond dissociation energies of host materials were predicted to be significantly small in radical ion species (*Chem. Mater.* **2016**, 28, 5791 and *J. Phys. Chem. C* **2016**, 120, 19987). These two groups suggested that suppression of radical ion species

could be the key approach to improving device lifetime. Note that our study is a significant advance from the previous studies, because it experimentally reveals that the radical species can be generated intermolecularly, even in the absence of charge injection. This finding is the first observation, and is fully consistent with the poor device longevity of blue-phosphorescent OLEDs. In addition, we focused on the molecular studies and revealed the kinetic processes of the generation and annihilation of radical ion pairs. We would like to emphasize that the kinetics results are in perfect accordance with the trend in device lifetime.

9. The authors compare L95 values for the four devices. Their plot makes this comparison impossible to see and is not the best way to evaluate these anyway. The authors should look at L70 or L50. Look at the inset to Figure 4e, it is impossible to see the L95.

Response and Revision: Following the reviewer's suggestion, we did re-draw Fig. 4e. The revised Fig. 4e now contains LT_{70} (See attached figure below).

10. The oxidation potential for the four dopants fall in the order $3 > 4 > 1, 2$, but the OLED lifetimes are $4 > 3 > 1, 2$. Their model says that the faster the back electron transfer is the longer the lifetime. Based on their numbers the fastest back electron transfer rate should be from 4, but 3 gives a clearly longer lifetime.

Response: This is misunderstanding of our data. The back electron transfer should occur in the Marcus-inverted region of electron transfer, because the driving forces are very large (2.63–2.94 eV). This prediction is in accordance with our experimental results. Fig. 3d demonstrates that the electron transfer is indeed located in the Marcus-inverted regime. Therefore, the reversal of ordering in the rate constant for back electron transfer adheres well to our mechanism: the greater driving force ($\text{Ir3} > \text{Ir4}$), the slower back electron transfer ($\text{Ir3} < \text{Ir4}$). In the case of **Ir1** and **Ir2**, they cannot be directly compared with the other two **Ir** dopants because they adhere to different Marcus plots. Finally, a linear relationship between the rate constant for back electron transfer and device lifetime further supported validity of our model (Fig. 4f).

Reviewer #1 (Remarks to the Author):

One of the important points I suggested is that the analyses were carried out in solutions, not in the devices. Unfortunately, this point has not been updated. The manuscript will become “outstanding” if the direct analyses in the devices have been carried out. However, not only myself but also all researchers related to this community will understand the difficulty of the analyses. The clarification of the degradation mechanism of blue OLEDs is very important and even without the analyses in the devices, this study provides new fruitful information about the degradation of the devices. The authors carefully and reasonably revised the manuscript on all the other issues. I now recommend the publication in Nature Communications as is.

Reviewer #2 (Remarks to the Author):

i have reviewed the authors' responses to my earlier comments and suggestions. I am satisfied with the manuscript revisions as well as the additional results that they provided in Supplementary Figure 19. I also appreciate their additional clarifications.

I recommend accepting the revised manuscript for publication in Nature Communications

Reviewer #3 (Remarks to the Author):

The authors have adequately addressed my concerns with their paper in the revised version.

Requests from the editorial office and our responses and changes:

1. *Nature Communications uses a transparent peer review system, where for manuscripts submitted from January 2016 we are publishing the reviewer comments to the authors and author rebuttal letters of our research articles online as a supplementary peer review file. Please let us know in the cover letter when submitting the final version of your manuscript if you wish to opt out of this scheme or not. If you are concerned about the release of confidential data, we do permit redactions in the interest of confidentiality. If you would like to remove such information from these files, then please let us know specifically what information you would like to have removed. Please note that we cannot incorporate redactions for other reasons. Reviewer names will be published in the peer review files if the reviewer comments to the authors are signed by the reviewer, or if reviewers explicitly agree to release their name. For more information, please refer to our FAQ page at: <https://media.nature.com/full/nature-assets/ncomms/authors/ncomms-transparent-peer-review.pdf>*

Response: We accept publication of all the communications of us and the reviewers during the processes of revisions. The materials do not contain any confidential data, so they are publishable as they are.

2. *Your manuscript should comply with our policies and format requirements, detailed in our checklist for authors at: http://www.nature.com/article-assets/npg/ncomms/authors/ncomms_manuscript_checklist.pdf*

Response: We have thoroughly examined the final manuscript, according to the Manuscript Checklist. Our manuscript fully complies with the policies and requirements. We attach a marked file of the Manuscript Checklist, along with this cover letter.

3. *Please also review the changes in the attached copy of your manuscript, which has been edited for style, and address the comments and queries I have added. If using Word, please use the 'track changes' feature to make the process of accepting your manuscript more efficient.*

Response and changes: We appreciate for the corrections and suggestions. We revised our manuscript and supplementary information, according to the comments provided the editor. Annotated versions of the files are being uploaded to enable 'track changes'.

4. *Data availability statements and data citations policy: All Nature Communications manuscripts must include a section titled "Data Availability" at the end of the Methods section or main text (if no Methods). For more information on this policy, and a list of examples, please see <http://www.nature.com/authors/policies/data/data-availability-statements-data-citations.pdf> In particular, the Data availability statement should include:*

- Accession codes for deposited data
- Other unique identifiers (such as DOIs and hyperlinks for any other datasets)
- At a minimum, a statement confirming that all relevant data are available from the authors
- If applicable, a statement regarding data available with restrictions

- If a dataset has a Digital Object Identifier (DOI) as its unique identifier, we strongly encourage including this in the Reference list and citing the dataset in the Data Availability Statement.

Response and changes: We included the section “Data Availability” at the end of the “Methods” in our revised manuscript:

“Data availability

The data that support the findings of this study are available from the corresponding author upon reasonable request.”

5. DATA SOURCES: We strongly encourage authors to deposit all new data associated with the paper in a persistent repository where they can be freely and enduringly accessed. We recommend submitting the data to discipline-specific, community-recognized repositories, where possible and a list of recommended repositories is provided here: <http://www.nature.com/sdata/policies/repositories>

If a community resource is unavailable, data can be submitted to generalist repositories such as figshare (<https://figshare.com/>) or Dryad Digital Repository (<http://datadryad.org/>). Please provide a unique identifier for the data (for example a DOI or a permanent URL) in the data availability statement, if possible. If the repository does not provide identifiers, we encourage authors to supply the search terms that will return the data. For data that have been obtained from publically available sources, please provide a URL and the specific data product name in the data availability statement. Data with a DOI should be further cited in the methods reference section. Please refer to our data policies here: <http://www.nature.com/authors/policies/availability.html>

Response: We do not deposit the data, because all the characterization data for the new materials are summarized in the Methods. Copies of the spectra are also included in the Supplementary Information. We indicated “Data Availability” to encourage readers to request data directly from us.

6. Please check whether your manuscript or Supplementary Information contain third-party images, such as figures from the literature, stock photos, clip art or commercial satellite and map data. We strongly discourage the use or adaptation of previously published images, but if this is unavoidable, please request the necessary rights documentation to re-use such material from the relevant copyright holders and return this to us when you submit your revised manuscript.

Response: We thoroughly examined our final manuscript and Supplementary Information whether they contain external images. They do not include any third-party images. All the graphs, figures and photos in our manuscript and Supplementary Information were produced by us.

7. Please see our requirements regarding characterization of structurally-novel chemical compounds, and the required format for compound characterization data: <http://www.nature.com/ncomms/journal-policies/editorial-publishing-policies#Characterization-materials>
Please note that this includes ¹H-NMR, ¹³C-NMR and high resolution mass spectrometry for all structurally-novel chemical compounds.

Response and changes: The structural characterization data (^1H and $^{13}\text{C}\{^1\text{H}\}$ NMR and HR MS data) of the compounds reported in our manuscript (the Ir series dopants and the H host) have been revised to meet the requirements of Nature Communications. All the changes are indicated in the annotated version of our manuscript.

8. We are committed to ensuring clarity and avoiding ambiguity in the mathematics in our papers. Consequently, please carefully check the mathematical terms throughout your manuscript (including labels on figures and figure captions) to ensure that it conforms strictly to the following guidelines. In mathematical terms, scalar variables (e.g. x , V , χ) should be typeset in italic, whereas multi-letter variables should be formatted without italic. Constants (e.g. \hbar , G , c) should be typeset in italics (the only exceptions being e , i , π , which should be typeset without italic) and vectors (such as r , the wavevector k , or the magnetic field vector B) should be typeset in bold without italics. In contrast, subscripts and superscripts should only be italicized if they too are variables or constants. Those that are labels (such as the 'c' in the critical temperature, T_c , the 'F' in the Fermi energy, E_F , or the 'crit' in the critical current, I_{crit}) should be typeset in roman. Please also ensure the same convention is followed in figure labels, axes, and such. Additionally, to avoid doubt, unit dimensions should be expressed using negative integers (e.g. $\text{kg m}^{-1} \text{s}^{-2}$ not kg/ms^2) or the word 'per'.

Response and changes: Our final manuscript and Supplementary Information have been revised to have mathematical representations, symbols, and units being consistent with the Styles of Nature Communications.

9. Your paper will be accompanied by a two-sentence editor's summary, of between 250-300 characters, when it is published on our homepage. Could you please approve the draft summary below or provide us with a suitably edited version.

"The short lifetime of blue-phosphorescent organic light-emitting devices owing to material degradation impedes their practical potential. Here, Kim et al. study the molecular mechanism of the degradation that involves exciton-mediated electron transfer as a key step for the generation of radical ion pairs."

Response and changes: We accept the editor's summary.

In addition to the changes listed above, we have moved Fig. 5 and the synthetic details of the Ir dopants to the main text from the supplementary information. All the figures have been revised to avoid use of both green and red colors.